# Identification and characterisation of vaginal bacteria-glycan interactions implicated in reproductive tract health and pregnancy outcomes

Virginia Tajadura-Ortega [1,2,3] ✉, Wengang Chai [1,2,3], Lauren A. Roberts [1,4], Yibing Zhang[1,2,3], Antonio Di Maio [2,3], Alexiane C. Decout [3,5], Benedita A. Pinheiro [6], Angelina S. Palma[6], Gian De Nicola [7], Lucia Riaposova[1,3], Belen Gimeno-Molina [1,3,8], Yun S. Lee[1,3], Hongzhi Cao [9], Vladimir Piskarev[10], Yukie Akune[2], Tiago R. D. Costa [11], Himani Amin [11], Lynne Sykes [1,3,8], Phillip R. Bennett [1,3], Julian R. Marchesi [1,4], Ten Feizi [1,2,3], Yan Liu [1,2,3] ✉ & David A. MacIntyre [1,3,12] ✉

*Lactobacillus* displacement from the vaginal microbiome associates with adverse health outcomes and is linked to increased risk of preterm birth. Glycans mediate bacterial adhesion events involved in colonisation and infection. Using customised glycan microarrays, we establish glycan interaction profiles of vaginal bacteria implicated in reproductive health. Glycan binding signatures of the opportunistic pathogens *Escherichia coli, Fusobacterium nucleatum* and *Streptococcus agalactiae* to oligomannose N-glycans, galactose-terminating glycans and hyaluronic acid, respectively are highly distinct from *Lactobacillus* commensals. Binding to sulphated glycosaminoglycans by vaginal bacteria is pH dependent, as is binding to neutral and sialic acid-terminating glycans by *F. nucleatum*. Adhesion of *Lactobacillus crispatus, Lactobacillus iners, Gardnerella vaginalis, S. agalactiae* and *F. nucleatum* to vaginal epithelial cells is partially mediated by chondroitin sulphate. *S. agalactiae* binding to chondroitin sulphate C oligosaccharides is inhibited by *L. crispatus*. This study highlights glycans as mediators of vaginal bacterial binding events involved in reproductive health and disease.

The vaginal microbiome is an important mediator of reproductive health and is a risk modifier of adverse pregnancy outcomes such as preterm birth[1–3]. Loss of *Lactobacillus* species and overgrowth of gram-negative bacteria are microbiological hallmarks of the common vaginal condition, bacterial vaginosis (BV)[4]. In pregnancy, *Lactobacillus crispatus* dominance of the cervicovaginal niche is protective against preterm birth[5–7], whereas *Lactobacillus iners* or a shift towards a high diversity composition is associated with increased risk of preterm birth[8,9]. This high diversity vaginal microbiota is often enriched in opportunistic, potentially pathogenic bacteria such as *Gardnerella vaginalis, Streptococcus agalactiae, Sneathia* and *Prevotella* species that promote reproductive tract inflammation[8,10]. *L. crispatus* prevents pathogen colonisation of the vaginal microbiome via several mechanisms including production of antimicrobial compounds and lactic acid, leading to low vaginal pH < 4.5 in most healthy asymptomatic women[11,12].

Glycans are made up of sugars arranged to form molecular epitopes specifically recognised by antibodies, lectins and carbohydrate binding proteins[13]. They form the bulk of the glycocalyx and are the main component of mucus in human secretions, including the cervicovaginal fluid (CVF)[14]. Glycans constitute the first point of contact in many cell-cell interactions, including host-microbiota interactions[8]. Host glycans modulate dynamics of microbial colonisation, serving as attachment and adhesion sites as well as nutrient sources[15–18]. Microbes can modify host glycans to avoid immune surveillance[19] during infection or induce immune tolerance through molecular mimicry[20,21]. There is increasing evidence that host glycan-bacteria interactions are important modifiers of microbiota composition in the lower reproductive tract, yet the knowledge on specific glycan structures involved is limited[17,18,20,22]. Glycan microarrays offer a powerful strategy to elucidate substrate specificity of glycan binding proteins, including those derived from microbes such as viruses, bacteria or parasites[23–26].

Here, we report the development of customised glycan microarrays for the detection and characterisation of the glycan binding specificities of key bacterial species from the vaginal microbiota implicated in reproductive health, like *L. crispatus, L. iners* and *G. vaginalis* and opportunistic pathogens, including *S. agalactiae, E. coli* and *F. nucleatum*, whose presence is associated with dysbiosis and adverse pregnancy and neonatal outcomes[15,27]. Our results reveal distinct glycan binding profiles by the opportunistic pathogens studied here, when compared collectively to commensal species within the vaginal microbiota and highlight pH-dependent interactions with sulphated glycosaminoglycans (GAGs) such as chondroitin sulphate (CS), a key mediator of bacterial adherence to vaginal epithelial cells.

## Results

### Glycan library and microarray design for bacterial binding studies

A library of 187 sequence-defined glycan probes (Supplementary Data 1) was constructed, and these were included in six different glycan microarray sets (Supplementary Fig. 1) used for the study. This library was designed to encompass representative glycans and glyco-epitopes from the glycocalyx and displayed on mucins, which form the major structural component of CVF[28] (Fig. 1). Glycans were converted into amino-terminating probes for their immobilisation on N-hydroxysuccinimide (NHS)-functionalised glass slides, producing covalent glycan microarrays. Mass spectrometry (MS) and, when necessary, NMR, were used for glycan probe characterization (Source Data File 1). As part of the microarray quality control (QC), the immobilisation and epitope integrity of the glycan probes used in this study were assessed using a collection of 33 sequence-specific glycan recognition systems (Supplementary Data 2), which exhibited the expected binding patterns with only a few exceptions (Source Data File 2) of additional or weak binding to related glycans not described before, and which are noted in the MIRAGE document under the Section of Glycan identification and quality control (Supplementary Data 3).

The well characterised binding of the pathogenic urinary tract infection (UTI)-associated strain *E. coli* C600 to oligomannose N-glycans was used to optimise conditions for bacterial binding[29]. Both fixed and live fluorescently labelled *E. coli* C600 bound specifically to oligomannose N-glycans on Microarray set 1 (Fig. 2A) with 69 glycan probes including N-, O-, blood group related, sialylated and N-acetylglucosamine (GlcNAc)-terminating glycans (Supplementary Fig. 2A). Consistent with previous findings of oligomannose binding by the type 1 pilus adhesin FimH of uropathogenic *E. coli*[30], strong binding to the N-glycan Man5 was observed across several independent experiments (Supplementary Fig. 2B)[29]. Concanavalin A, a plant lectin with known high avidity for α-linked mannose[31], used as control in the same assay, showed strong binding to oligomannose N-glycans and weak binding to the complex-type N-glycan NA2. Glycan binding of three other *E. coli* strains (*E. coli* 789, 900 and 901) isolated from the

urine of UTI patients[32] was also tested (Fig. 2B, C). *E. coli* 789 and 901 strains strongly bound oligomannose N-glycans. Binding by *E. coli* 900 to oligomannose N-glycans was comparatively weak. Electron microscopy (EM) analysis of the *E. coli* isolates revealed that *E. coli* 900 displays substantially shorter and fewer fimbriae compared with the three other strains tested (Fig. 2D).

### Comparison of glycan binding profiles of vaginal commensals and pathogens

We next performed glycan binding profiling of key vaginal commensals such as *L. crispatus* and *L. iners*, and potentially pathogenic bacteria including *G. vaginalis, S. agalactiae, E. coli* and *F. nucleatum*. Binding experiments were conducted on a total of 22 strains, 17 of which were patient-derived isolates[32] (Supplementary Data 4) at mild acidic (pH 4) and neutral pH (pH 7). This was to reflect the pH gradient within the female reproductive tract which ranges from pH <4.5 in the lower vagina, to pH 6.7-7.2 in the cervix and pH 7.5-8.0 in the upper uterine cavity[11]. We first examined glycan binding to a set of 90 neutral and sialylated probes on Microarray set 2 (Probes #1-90; Supplementary Data 1). Glycan binding footprints are summarised in Fig. 3, with representative strain-level binding for each species presented in Supplementary Fig. 3. Bacterial binding was not observed to any of the short glycan probes with an amino-propyl (-sp) linker (probes #91-110 on Microarray set 3; Supplementary Data 1), despite showing expected binding with lectins and antibodies in QC tests. Oligomannose N-glycan binding by fimbriated *E. coli* strains was highly specific, a feature not shared by other species tested (Fig. 3 and Supplementary Fig. 3). While almost all species showed strong binding profiles at pH 4 and pH 7 to a subset of neutral and sialylated glycans, *F. nucleatum* 23726 strongly bound fucosylated, sialylated and galactose (Gal)-terminating glycans only at neutral pH, emphasising the significance of pH for this bacterium (Supplementary Figs. 3B, 4 and Fig. 4). Mucin O-glycans (#1, #2, #3 and #5), Gal-terminating poly-N-acetyllactosamine (polyLacNAc) glycans (#27-29, #30-39), including glycans with internal blood group epitopes, and non-sulphated chondroitin oligosaccharides (#88-90) were specifically bound by *F. nucleatum* 23627 (Supplementary Fig. 3). This result was validated using on-array inhibition assays which showed a minimum 60% decrease in binding by *F. nucleatum* 23627 to all Gal- and N-acetylgalactosamine (GalNAc)-terminating glycans in the presence of free Gal, but not free glucose (Glc) (Fig. 4A). Strong binding by *F. nucleatum* 23726 to α(2-6) sialic acid-terminating glycans (#5, #13-15, #24 and 26) and several linear and branched glycans containing the blood group H type-1 epitope (#54, #79-82) was also observed. Interestingly, binding to the branched α(2-6) sialyl core-1 O-glycan (#5), the α(2-6) sialylated glycan LSTb (#24) and the blood group terminating glycan (#78), which also contain a Gal-terminating antenna, was inhibited by free Gal, whereas binding to linear α(2-6) sialic acid-capped polyLacNAc (#26) or sialyl N-glycans (#13-15), was not. Binding of *F. nucleatum* to epithelial cells has been shown to be mediated by the autotransporter and Gal-inhibitable adhesin, Fap2[27]. Deletion of *fap2* abrogated *F. nucleatum* 23726 binding to Gal-terminating glycans, blood group terminating glycans and O-glycans. However, significant binding remained to sialic acid-terminating glycans from fetuin (Fig. 4B).

### Vaginal bacteria bind blood group A, B, H and Lewis active glycans as well as sialylated glycans

Previous studies have reported high levels of fucosylation and sialylation in the N-and O-glycans of CVF from pregnant and non-pregnant women[28,33]. Moreover, expression of specific blood group epitopes including A, B, H(O), Lewis a, Lewis x and Lewis y have been reported in CVF[33]. We observed binding by *L. crispatus, L. iners, S. agalactiae, G. vaginalis* and *E. coli* strains to blood group-terminating glycans with A, B, H and Lewis epitopes on linear type 1, type 2 or type 4 polyLacNAc

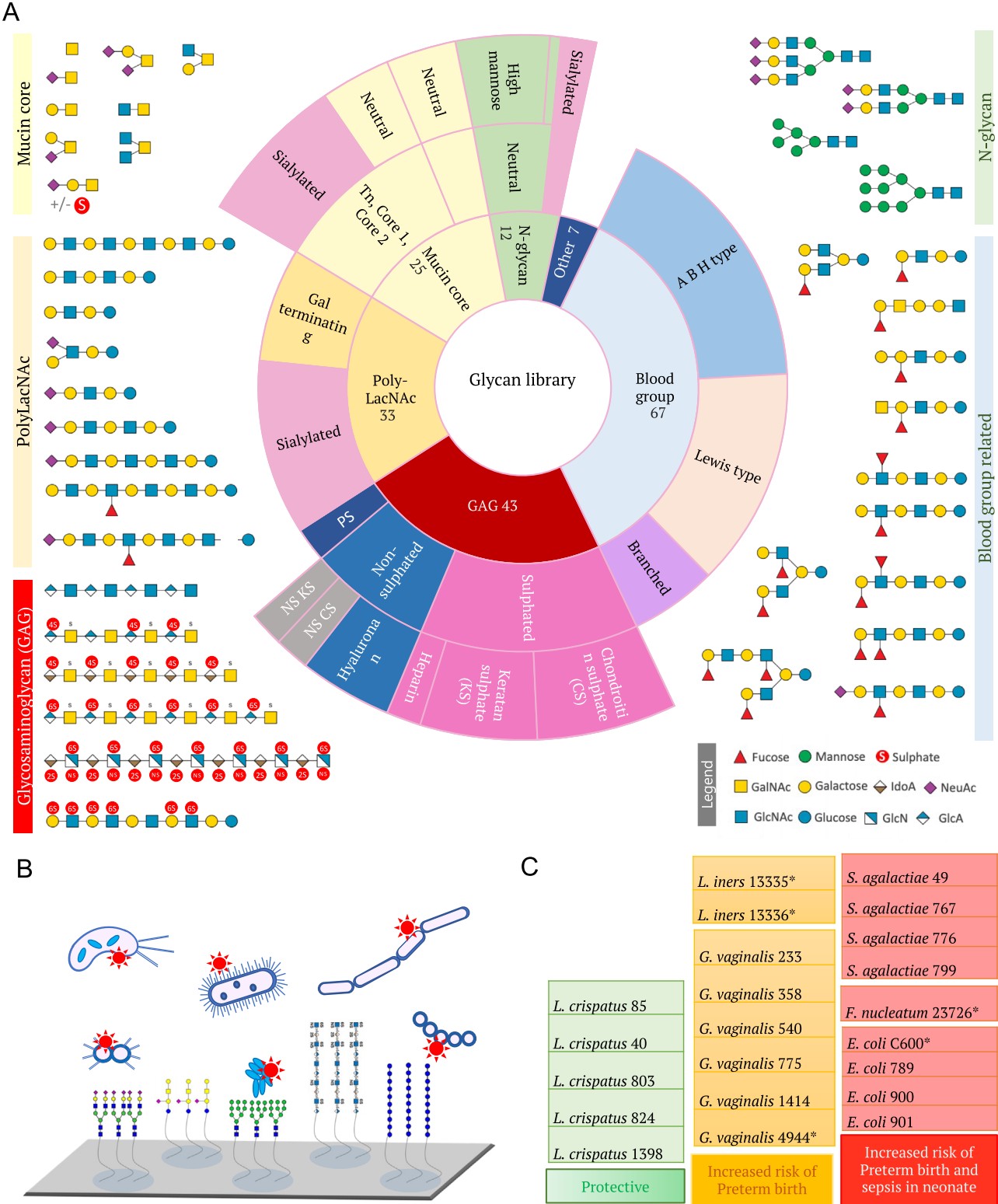

**Fig. 1 | Overview of the sequence-defined glycan library used in this study. A** Pie chart representing the glycan categories included in a customised sequence-defined glycan library for binding studies of vaginal microbiota. Numbers indicate the number of probes in main categories (Supplementary Data 1). Examples of glycans in each category are shown. PS polysaccharide, NS non-sulphated. **B** Diagram representing the methodology used for analysis of whole bacteria binding to sequence-defined glycans on microarrays. Glycans are covalently immobilised on NHS glass slides and binding performed with fluorescently labelled bacterial cultures of vaginal microbiota. **C** Bacterial collection with strain specification used for the study (Supplementary Data 4). * Strains purchased from ATCC or DMZ collections.

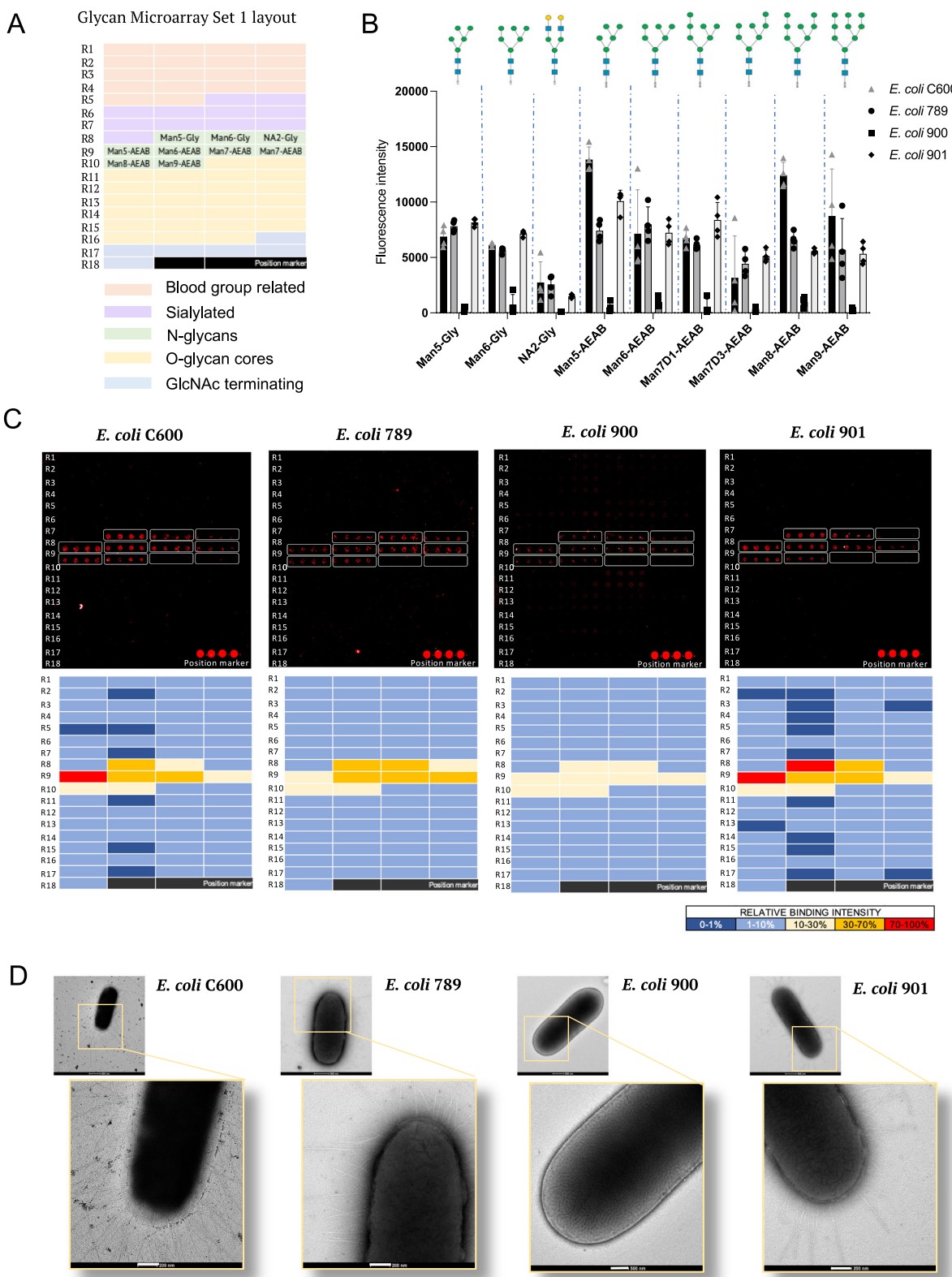

chains or branched human milk oligosaccharides (HMOs) (Fig. 3 and Supplementary Fig. 3), but not to the Gal-terminating type 1 or type 2 backbones (#27-34). Binding to blood group glycans A, B, H, Lewis x and Lewis b (#42, #45, #47, #49, #55, #60, #71, #82) was consistent across all *L. crispatus* strains tested at both pH 4 and pH 7, whereas for *L. iners, G. vaginalis* and *S. agalactiae*, binding was both pH and strain dependent (Fig. 3), except for binding to the HMO MFLNH with a terminal blood group H epitope (#82) by *G. vaginalis*, which was

observed for all strains tested. No binding was detected by *L. iners* 13335 at neutral pH (Supplementary Fig. 3), again highlighting species-dependent pH mediation of glycan binding events. At strain level, differences were also observed between clinically derived *E. coli* strains (789, 900, and 901) compared to the ATCC C600 strain. *S. agalactiae* did not bind to the linear LNFP-I probe #42 but did bind strongly to the bi-antennary HMO glycans (#82 and 83) containing the LNFP-I sequence (Fig. 3)[34]. All bacterial strains exhibited strong binding to

**Fig. 2 | Comparison of glycan binding on glycan microarrays and electron microscopy of *E. coli* strains isolated from UTI patients. A** Printing layout of glycan Microarray set 1 indicating the location of the main glycan categories for the 69 sequence-defined glycan probes included in the microarray (Supplementary Data 1). **B** Bar charts showing the mean fluorescence intensities of bacterial binding signals to glycans printed on quadruplicated spots on the glycan Microarray set 1. Error bars represent the SD of quadruplicates for each glycan probe on the microarray (*n* = 1). Assay is representative of three independent experiments. Structures of bound glycans are shown. Fixed cells were used. Source data are provided as a Source Data File. **C** Top: Microarray images of fixed fluorescently labelled bacteria binding to glycans on Microarray set 1. Bottom: Heatmaps of relative binding strengths amongst the four *E. coli* strains tested to glycans on Microarray set 1. Colour code: Dark blue (0–1 %); Light blue (1–10 %); Yellow (10–30 %); Orange (30–70%); Red (70–100%). 100%, the maximum binding score observed across the four experiments. **D** Electron-microscopy images of whole bacterial cells of *E. coli* strains C600, 789, 900, and 901. Enlargements showing differences in fimbriae sizes. White bars in enlargements indicate 200 nm. Images are representative of two independent experiments.

α(2-3) or α(2-6) sialic acid-terminating polyLacNAc glycans apart from *E. coli* C600 (Fig. 3). Weak, yet consistent, binding to sialylated N-glycans containing a mixture of α(2-3) and α(2-6) sialyl linkages was observed for most *L. crispatus* strains tested and some strains of *L. iners*, *G. vaginalis* and *S. agalactiae* (Fig. 3 and Supplementary Fig. 3).

## Vaginal bacteria show pH dependent binding to CS and heparin
We next investigated the capacity of vaginal bacteria to bind GAGs, which have been shown to be abundantly expressed on the cell surface and in the extracellular matrix of endometrial and cervico-vaginal epithelia[35–37]. To test bacterial binding to GAGs, Microarray set 4 containing 36 GAG oligosaccharides representing different chain lengths of chondroitin sulphate A (CSA), CSB (dermatan sulphate), CSC, keratan sulphate (KS), heparin (a highly sulfated form of HS), and hyaluronic acid (HA) was constructed (Supplementary Data 1). At pH 4, most strains bound CS, heparin and KS oligosaccharides, but comparatively weak or no binding was observed to the non-sulfated KS probes (Fig. 5A). Of the *E. coli* strains tested, only C600 and 901 demonstrated weak binding to oligosaccharides derived from CSs, heparin and KS. Binding to CSs and heparin was pH dependent for most bacterial strains tested. At pH 7, only *F. nucleatum* and some *Lactobacillus* strains bound strongly to CSA or heparin oligosaccharides, respectively (Fig. 5A). The pH-dependent binding of bacteria to CSs and heparin oligosaccharides was observed in both HEPES and acetate buffers (Supplementary Fig. 5A).

Binding to HA was restricted to a few strains of *S. agalactiae* and *G. vaginalis*. HA binding by *S. agalactiae* has been previously reported[38–40]. We identified in the urogenital derived strains of *S. agalactiae* 49, 767 and 776 a putative hyaluronidase with an appended carbohydrate binding module (CBM), that shares significant homology to the CBM70 domain of the *S. pneumoniae* hyaluronidase that targets HA[41,42]. Consistent with microarray binding data of whole bacteria, Isothermal Titration Calorimetry (ITC) demonstrated that a 14-mer HA oligosaccharide is a substrate for the recombinant full-length *S. agalactiae* 767 hyaluronidase (*Sa* Hyal_767) (Supplementary Fig. 6A, B). No enzymatic reaction was observed with heparin 14-mer or the negative control LNnT tetrasaccharide (Supplementary Fig. 6C).

The interactions of CS and heparin oligosaccharides with *L. crispatus* 1398, *L. iners* 13335, *G. vaginalis* 775, *S. agalactiae* 776, *E. coli* C600 and *F. nucleatum* 23726 were further investigated with biotinylated CSA, CSB, CSC and heparin polysaccharides immobilised on NHS functionalized glass slides on Microarray set 5 (Supplementary Fig. 7A). Again, strong pH-dependent binding to all GAGs was observed for *L. crispatus* 1398, *G. vaginalis* 775, *S. agalactiae* 776 and *F. nucleatum* 23726; whereas *L. iners* only showed weak binding signal in HEPES buffer (Fig. 5B, C and Supplementary Fig. 5B). *F. nucleatum* 23726 gave strong binding to CSA, which is enriched in endometrial tissue and placenta[43,44], and heparin polysaccharides at pH 7 (Supplementary Fig. 7B–D). *E. coli* C600 did not bind GAG polysaccharides in any of the conditions tested.

## Binding of *L. crispatus*, *L. iners*, *S. agalactiae* and *F. nucleatum* to vaginal epithelial cells is partially mediated by CS
Using flow cytometry, we observed that cultured vaginal epithelial cells (VK2) demonstrate high surface expression levels of CSs and heparan sulphate (HS) but not HA or KS (Supplementary Fig. 8A). Treatment of VK2 cells with chondroitinase ABC resulted in a significant decrease in anti-CS56 binding (Supplementary Fig. 8B). Binding of vaginal epithelial cells to fluorescently labelled *L. crispatus* 1398, *L. iners* 13336, *G. vaginalis* 775, *S. agalactiae* 776 and *F. nucleatum* 23726 was observed at pH 4 and pH 7, however chondroitinase ABC treatment significantly reduced bacterial binding at pH 4 but not at pH 7 (Fig. 6A and Supplementary Fig. 8C). Consistent with this, pre-incubation of *L. crispatus* with CSC polysaccharide significantly reduced bacterial binding to VK2 cells at pH 4 (Fig. 6B). Treatment with heparinase to remove HS did not affect *L. crispatus* 1398 or *G. vaginalis* 775 binding (Supplementary Fig. 8D). As several studies have indicated that *L. crispatus* may inhibit *S. agalactiae* colonisation in the vaginal niche[45–48], we next performed on array competition assays, where binding of *S. agalactiae* 776 to CSC 14-mer was assessed in the presence of increasing amounts of *L. crispatus* 1398. Binding of *S. agalactiae* 776 to CSC 14-mer decreased in a dose-dependent manner with increasing concentrations of *L. crispatus* (Fig. 6C), indicating that *L. crispatus* can compete with *S. agalactiae* for CSC binding.

## Discussion
Using customised glycan microarrays, we unveiled a diverse range of glycan structures recognised by vaginal bacteria implicated in reproductive health and disease states. These include GAGs that decorate plasma membrane and extracellular matrix glycoproteins, neutral and sialylated glycans carried in glycolipids at the plasma membrane, and N- and O-glycans presented on membrane-bound and secreted glycoproteins like mucins, the major protein constituents of CVF. We identified glycans specifically recognised by the opportunistic pathogens *E. coli*, *F. nucleatum* and *S. agalactiae*, that are not bound by commensals, whereas other glycan binding activities were shared by both beneficial and potentially pathogenic bacteria. Further, we demonstrated that adhesion of bacteria to vaginal epithelial cells is partially glycan mediated.

The utility of our glycan microarray approach was first tested using four *E. coli* strains. *E. coli* strain 900 demonstrated reduced oligomannose glycan binding compared to the other strains tested. Oligomannose binding by *E. coli* is known to be mediated by the adhesin, FimH1, which is located at the tip of the fimbria[30]. Consistent with our microarray results, electron microscopy showed that *E. coli* strain 900 had comparably shorter and fewer fimbria, despite being cultured in similar conditions. The abnormally short and reduced number of cellular appendices in *E. coli* 900 is likely to be abrogating oligomannose binding in this strain. These results thus highlight the capacity of glycan microarray studies to effectively discern binding characteristics attributable to different bacterial strain morpho-phenotypes.

Glycan microarray technologies, since first introduced in early 2000[49,50], have revolutionized the molecular dissection of glycan-protein interactions in a miniaturised and high-throughput fashion. Significant advances have been made in applying glycan microarrays to analyze the glycan-binding specificities of microbial glycan-binding proteins, including adhesins (lectins) from viruses, fungi, and bacteria, bacterial toxins and CBMs, as well as whole virus particles. However, the application of this technology to whole bacterial cell binding has remained relatively limited. A few studies have utilised glycan array

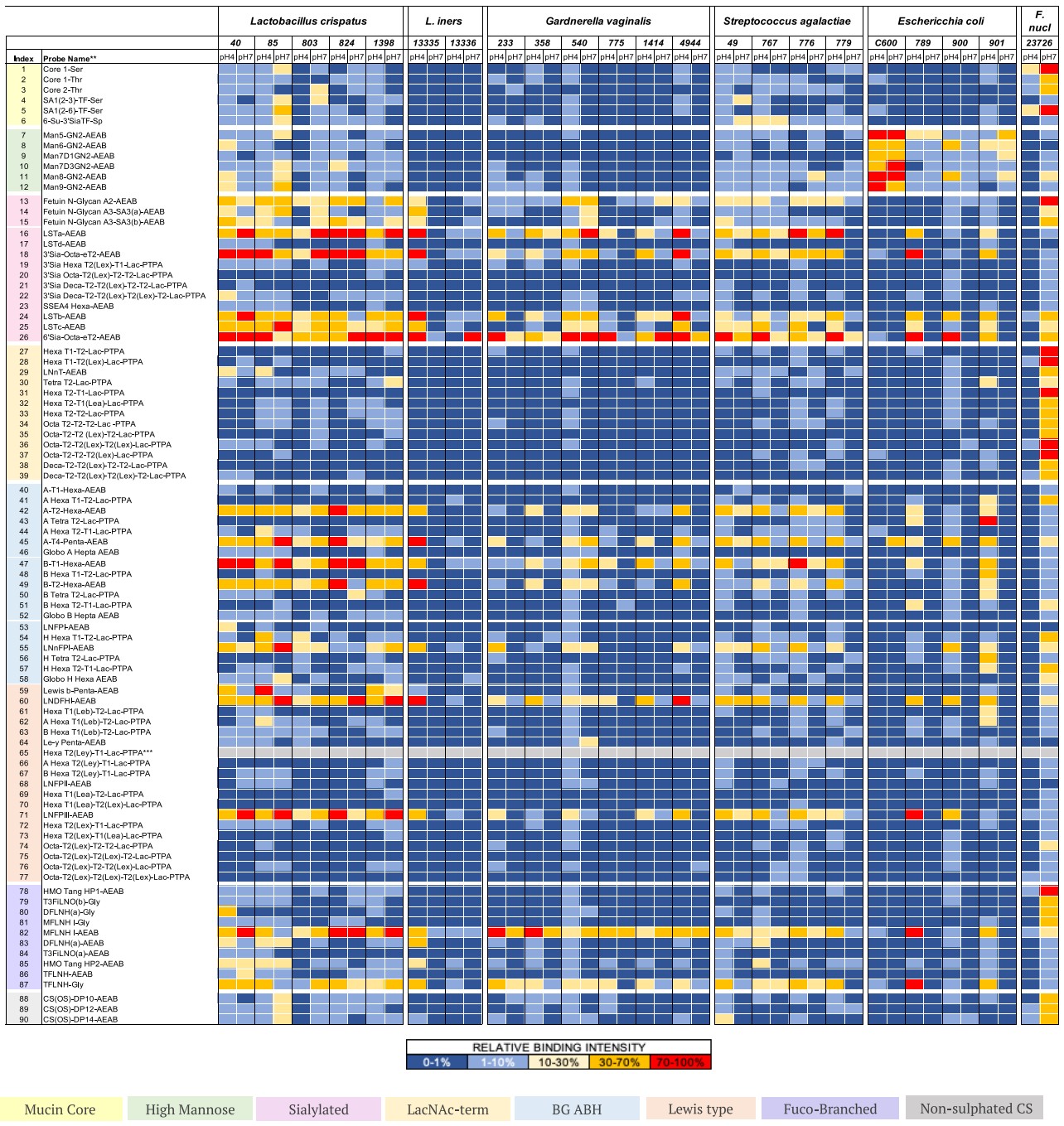

**Fig. 3 | Binding of bacteria from the vaginal microbiota to neutral and sialic acid terminating glycans on glycan microarrays.** Heatmap showing the relative binding intensities of live fluorescently labelled bacterial cultures of *L. crispatus*, *L. iners*, *G. vaginalis*, *S. agalactiae*, and fixed *E. coli* and *F. nucleatum* strains to 90 neutral and sialic acid terminating glycans at pH 4 and pH 7 on Microarray set 2 with glycans printed at high density (300 μM). Colour code: Dark blue (0–1%); Light blue (1–10%); Yellow (10–30%); Orange (30–70%); Red (70–100%). 100%, the maximum binding score observed across pH 4 and pH 7 for a given strain. Glycan probes are grouped according to their sequences into categories and annotated by the coloured panels defined at the bottom of the figure. Histograms of representative strains are shown in Supplementary Fig. 3. Grey cells correspond to glycan probe #65 that did not pass quality control in this set.

analysis to investigate host glycan binding by whole bacterial cells of pathogenic bacteria, including *Campylobacter jejuni*[51], Group A Streptococcus[52] and *Neisseria meningitidis*[53,54]. These studies have provided valuable insights into bacteria-glycan interactions, but also present several limitations. For example, many of these studies examine only a limited selection of bacterial species and strains, and the glycan libraries used are often narrow in scope, being restricted to commercial glycans or exclusively sialylated glycans. The conclusions drawn from these studies are often limited to monosaccharide specificity and less quantitative. To our knowledge, this is the first time that

GAG oligosaccharides of different types and chain lengths have been interrogated for bacterial cell binding side by side with neutral and sialylated glycans of diverse types.

During assay condition optimisation, fucose-dependent blood group binding was detectable only at a high glycan printing concentration (300 μM), likely due to increased glycan density. In contrast, strong binding of anti-carbohydrate antibodies was observed even at lower concentrations (50–100 μM) on the same array (Supplementary Fig. 9). This suggests an overall weak binding avidity of bacterial cells to the blood group glycans presented on the array. Thus, for screening

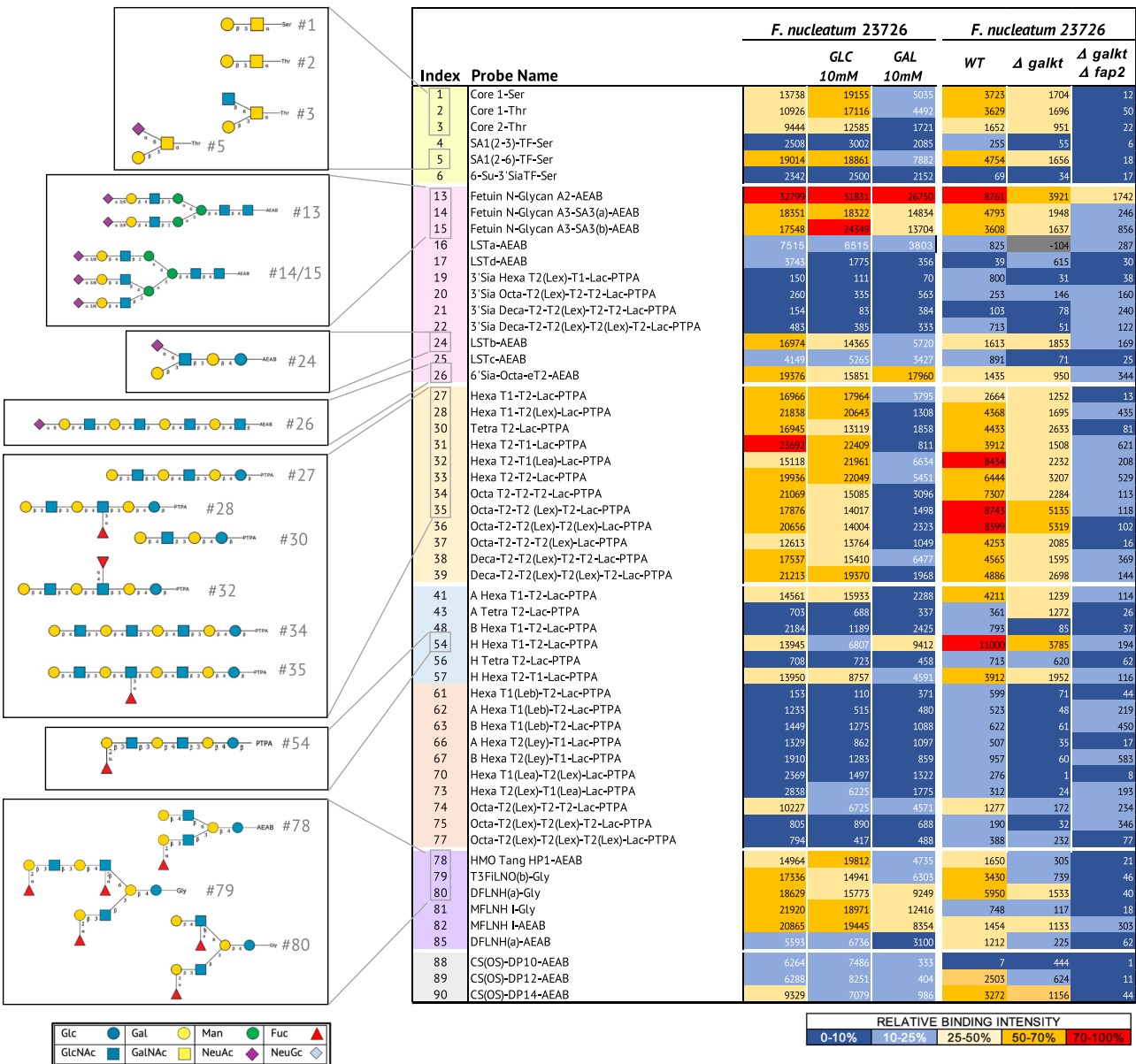

**Fig. 4 | Glycan binding studies of *wt and Δfap2 Fusobacterium nucleatum* with sequence-defined covalent microarrays reveals binding to sialylated, fucosylated and galactose-terminating glycans.** (Left panels) Heatmaps of relative binding strengths of fixed fluorescently labelled *F. nucleatum* 23726 to neutral and sialylated glycans at pH7 in the presence of 10 mM galactose, 10 mM glucose or buffer only. Source data are provided in Source Data File 5. (Right panels) Heatmaps of relative binding strengths of fixed fluorescently labelled *wt, Δgalkt and Δgalkt Δfap2 F. nucleatum* 23726 to neutral and sialylated glycans at pH7. Values on heatmap indicate the mean fluorescence intensities of quadruplicate spots on Microarray set 2a and 2b. Colour code for binding intensities: Dark blue (0–10%); Light blue (10–25%); Yellow (25–50%); Orange (50–70%); Red (70–100%). 100%, the maximum binding score observed across the three conditions tested. Glycan structures corresponding to selected bound glycan probes are shown. Glycans categories are coloured as follows: Yellow: Mucin core related; Pink: Sialylated; Orange: LacNAc terminating; Blue: Blood group A, B, H glycans; Peach: Lewis blood group related; Purple: Fucosylated branched glycans from human milk; Grey: Non-sulphated chondroitin sulphate. * (-104) Grey cells were flagged for artefact/background on slide.

our vaginal microbiota collection, a printing concentration of 300 μM was used for Microarray set 2. With the exception of *F. nucleatum*, we observe that most strains bound to neutral glycans derivatised with the AEAB (N-(aminoethyl)-2-aminobenzamide) linker (Fig. 3), suggesting that presentation and the chemical environment around the glycan, are likely to be important for whole bacterial binding to glycans[55,56]. Again, this effect was not observed in antibody and plant lectin binding (Source Data File 2).

Bacteria express multiple glycan-binding proteins (GBPs), including lectins and CBMs, which can simultaneously mediate interactions with host glycans, and the subcellular localisation of these GBPs varies (e.g., secreted vs membrane-bound). This diversity likely

explains our observation of bacteria binding to multiple glycan types with varying signal intensities. Our recent comparative genomics study revealed a wider repertoire of GBPs in vaginal pathogens and pathobionts compared to commensal species[42]. Except for a few glycans, the binding profiles of commensals *L. crispatus* and *L. iners* and potentially pathogenic bacteria like *G. vaginalis* and *S. agalactiae*, observed here were overlapping. This might reflect competition for similar glycans as described for GAG binding, but we cannot rule out that modification and utilisation of glycans, which involves a wide repertoire of additional enzymes involved in transport and processing of glycans inside the bacteria[57,58], might also differ from commensals to potentially pathogenic bacteria. Further glycan-

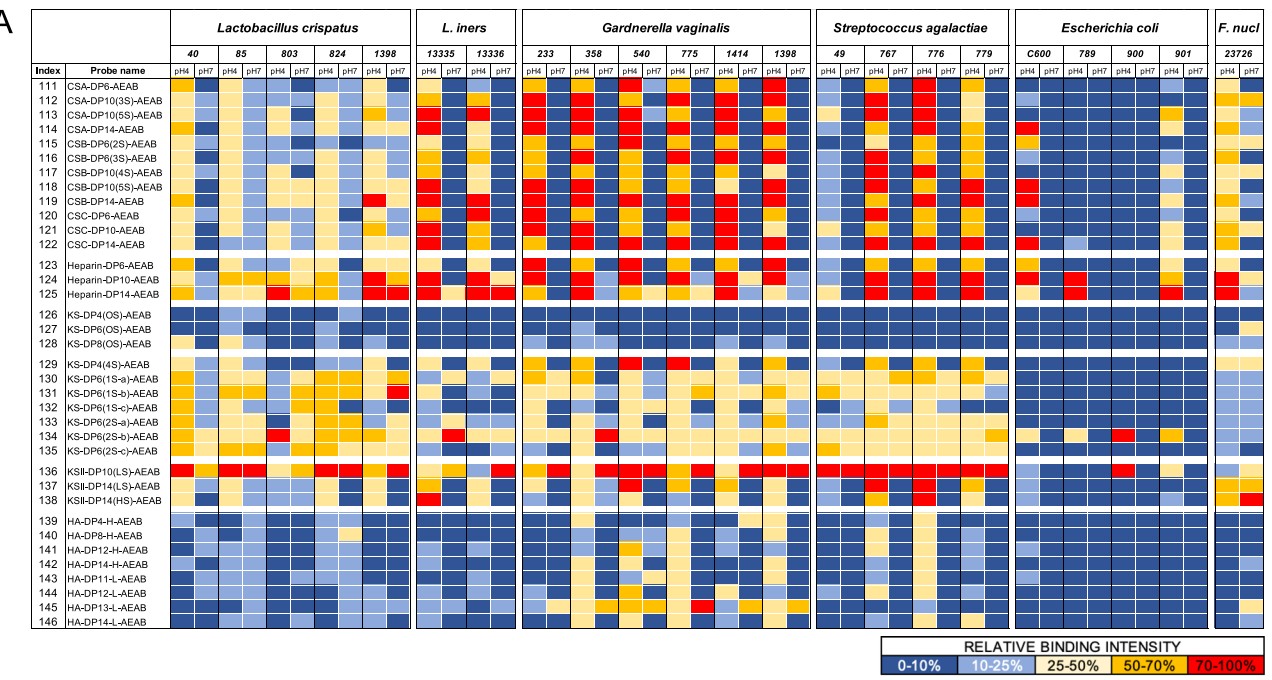

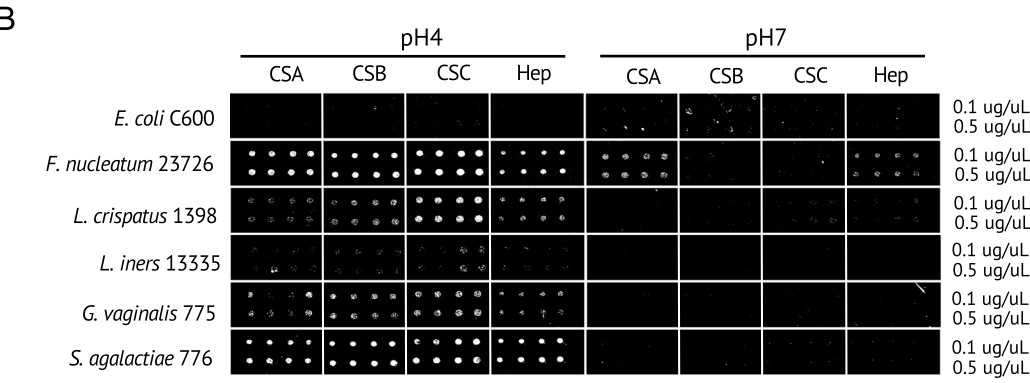

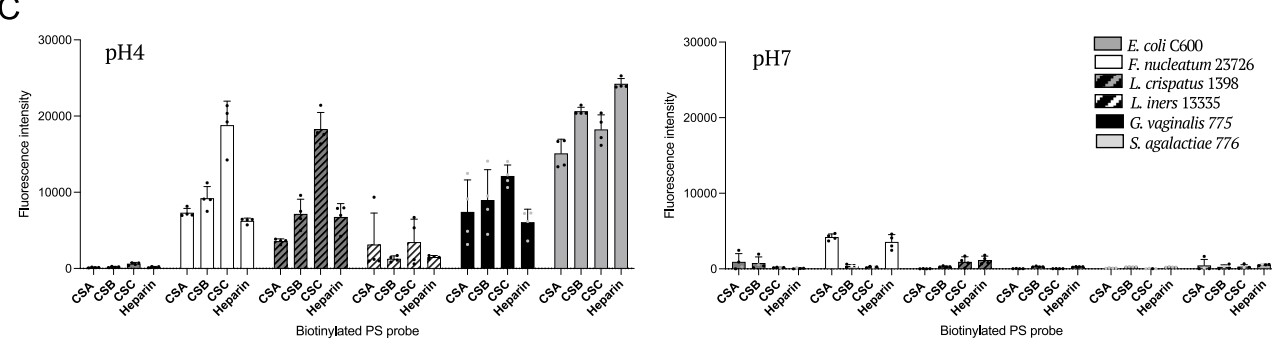

**Fig. 5 | Glycosaminoglycan binding by bacteria of the vaginal microbiota.**
**A** Heatmap showing the relative binding intensities to 36 GAG oligosaccharides at pH4 and pH7 of fixed fluorescently labelled strains of *E. coli and F. nucleatum*, and live *L. crispatus, L. iners, G. vaginalis and S. agalactiae*. The average rank of the mean fluorescence intensities from at least two independent experiments on Microarray set 5 was calculated (Source Data file 6) and the resulting values coloured as follows: Dark blue (0–10 %); Light blue (10–25 %); Yellow (25–50 %); Orange (50–70 %); Red (70–100%). 100%, the maximum binding score observed across pH4 and pH7 for a given strain. **B** Glycan microarray images of fluorescently labelled *E. coli* C600*, F. nucleatum* 23726*, L. crispatus* 1398*, L. iners* 13335*, G. vaginalis* 775 and *S. agalactiae* 776 binding to GAG polysaccharides at pH4 or pH7. Images are representative of three independent experiments. **C** Bar charts showing the mean fluorescence intensities of bacterial binding at pH4 or pH7 to GAG polysaccharides on Microarray set 6 printed at 0.5 mg/ml. Error bars represent SD of quadruplicate spots for each polysaccharide on the microarray (*n* = 1). Assay is representative of three independent experiments. Source data are provided in Source Data File 7.

binding activities may also exist that were not captured by the approaches used in this study.

Our GAG array results highlighted strong binding of *L. crispatus, L. iners, G. vaginalis, F. nucleatum* and *S. agalactiae* to the sulphated

GAGs, CSs and Heparin, which was largely pH dependent. Binding of *E. coli* to the same GAGs was comparatively weak. *S. agalactiae* and some strains of *G. vaginalis* also demonstrated strong binding to the non-sulphated GAG, HA. These findings likely reflect the differing capacity

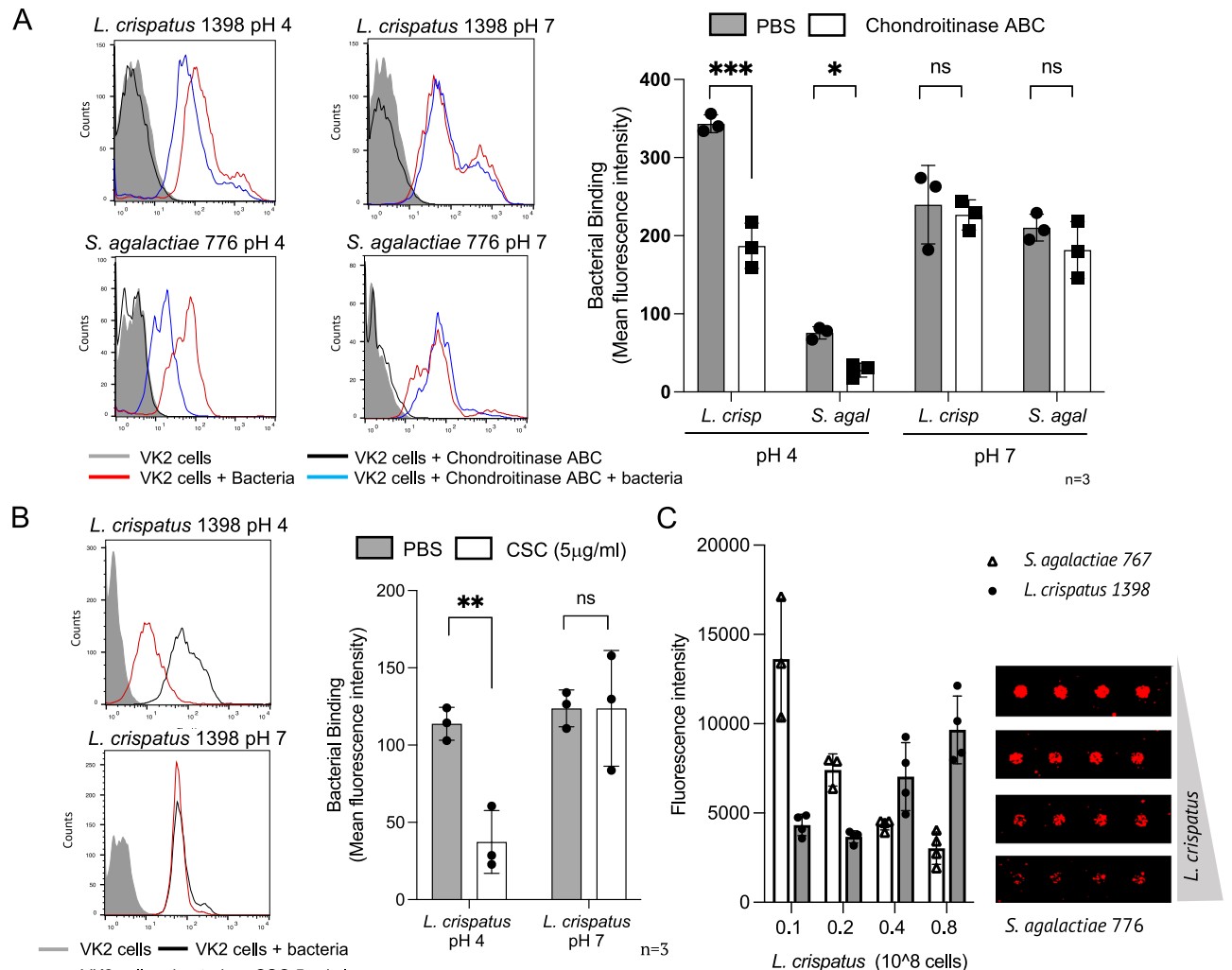

**Fig. 6 | Analysis of bacterial binding to VK2 vaginal epithelial cells and glycan binding competition by bacteria on glycan microarrays. A** Flow cytometry-based analysis of live bacterial binding to VK2 cells treated or not with chondroitinase ABC. Bar charts show the mean fluorescence intensities of the bound bacteria at pH4 and pH7. Error bars represent SD of three independent experiments for each condition ($n = 3$). **B** Flow cytometry-based analysis of the binding of *L. crispatus* preincubated or not with CSC polysaccharide to VK2 cells. Bar charts show the mean fluorescence intensities of the bound bacteria at pH4 and pH7. Error bars represent SD of three independent experiments for each condition ($n = 3$).

**C** Left: Bar charts show the mean fluorescence intensities of the binding at pH4 of fluorescently labelled *S. agalactiae* 776 and *L. crispatus* 1398 to a CSC 14-mer oligosaccharide in the presence of increasing amounts of *L. crispatus* 1398. Error bars represent SD of triplicates on the microarray for each glycan ($n = 1$). One experiment representative of two independent experiments is shown. Right: Representative images of *S. agalactiae* 776 cells binding to CSC 14-mer on the microarray in the presence of increasing amounts of *L. crispatus* 1398. *** $p < 0.0001$; *$p < 0.039$; ns: non-significant; 2-way ANOVA (A) and **$p < 0.0067$; ns: non-significant B). Source data are provided in the Source Data File 7.

of vaginal bacteria to adapt to dynamic changes in pH and adhere to different glycan structures encountered within the reproductive tract[35,37]. For example, during pregnancy, HA expression is highly increased in the cervix, whereas under-sulphated CSA is enriched in the neutral or slightly basic environment of the placenta[11,43,59]. The GAG binding characteristics of *F. nucleatum* and *S. agalactiae* described here are consistent with their capacity to colonise both the lower and upper reproductive tract in pregnancy, where pH is higher than in the vagina, which could potentially lead to infection and increased risk of preterm birth and neonatal sepsis[60–62].

Several studies have linked *F. nucleatum* colonisation of the cervicovaginal niche during pregnancy with increased risk of preterm birth and subsequent risk of early onset neonatal sepsis[63,64]. Amniotic fluid infection has also been previously linked to Fusobacteria colonisation of the vagina[65]. Our data provides evidence of direct and specific recognition by *F. nucleatum* of glycans terminating in Gal and GalNAc, such as the mucin core-1 O-glycan (Galb1-3GalNAc), non-sulphated chondroitin and several polyLacNAc glycans. Binding to Gal,

GalNAc and blood group glycans, but not sialylated glycans was abolished in the *fap2* deletion mutant. These results are consistent with previous studies using indirect methods like galactose or lectin competition/ inhibition of bacterial binding to cell lines and tissues, including placental epithelial cells, that reported *F. nucleatum* binds to Gal-terminating glycans (including the mucin core 1 O-glycan) through the autotransporter Fap2[27]. Gal-terminating LacNAc chains targeted by *Fusobacterium* are present in O-glycans from CVF[66], which may provide a potential explanation for *Fusobacterium* adhesion and colonisation of the vaginal niche[65]. Binding to sialic acid terminating N-glycans, also present in CVF, was observed in a *fap2* deletion mutant suggesting that other adhesins mediate sialic acid binding by *F. nucleatum*[67,68].

In the lower reproductive tract, a decrease in sialylation of polyfucosylated N-glycans in cervicovaginal fluid is associated with the presence of high diversity CST-IV community state type[33], which includes known sialic acid foragers like *Gardnerella*[18,69]. Here, using glycan arrays, we show that vaginal species including *L. crispatus, L. iners, G. vaginalis* and *S. agalactiae* have the capacity to bind several

blood groups and sialic acid-terminating glycans (N- and O-glycans) present in CVF[33]. Notably, binding to these non-GAG glycans was weaker compared to the binding to GAG sequences. Detectable signals required higher glycan probe concentrations, suggesting a need for increased surface density. Interestingly, except for *L. crispatus*, condition and strain-specific variations in neutral and sialic acid glycan binding were observed. Subspecies and strain to strain variability of *G. vaginalis* have been shown to differ in terms of virulence factors expressed, biofilm formation capacity and/or variability on the mucosal immune responses elicited[70–72]. Such variations could underlie differing functional abilities, including glycan binding as observed in this study, with exposure to different lectins, capsule or cell wall components. Consistent with these findings, specific sub-species or strains of *G. vaginalis* have recently been implicated in increased risk of preterm birth[5,73,74]. Despite this, consistent binding to certain glycans like H type 1 containing human milk oligosaccharides was observed.

Phenotypic differences in host blood group glycan expression have been correlated to host disease susceptibility by human pathogens such as enterotoxigenic *E. coli* or *Helicobacter pylori*, which express adhesins targeting a specific blood group epitope[75]. In addition, blood group B individuals are more susceptible to *E. coli* and *Group B Streptococcus* infections[76,77]. Histo blood group antigen (HBGA) expression can be used in patient risk stratification[78,79]. In the gut, blood group glycans are used by commensal bacteria as carbon sources thus shaping microbiota composition, which has implications for chronic inflammatory bowel disease[57,80,81]. Probiotic lactic acid-producing bacteria like *Levilactobacillus brevis* and *L. gasseri* have also been reported to bind to blood group antigens that might help with colonisation[82]. Blood group glycan expression in human epithelial cells and secretions such as saliva, breast milk or mucus depends on the secretor gene (FUT2). The FUT2 α-1,2-fucosyltransferase enables synthesis of the H(O) antigen, which can be elongated by either a GalNAc-transferase or a galactosyl-transferase to produce A and B antigens, respectively. Our findings that vaginal pathogens and commensals bind to blood group glycans will inform follow up studies on blood group glycan binding and utilisation by vaginal microbiota to provide mechanistic insight into recent reports of increased preterm birth risk in non-secretor individuals harbouring pathogen enriched vaginal microbiomes[83]. Further studies investigating the relationship of microbiota composition, HBGA expression and blood group antigen binding and utilisation by bacteria in pathogen-driven disease, particularly in the context of reproductive tract infections, are also warranted.

Our study also demonstrated that *L. crispatus* competes for binding to specific glycan structures targeted by vaginal pathogens, such as *S. agalactiae*, in a pH dependent manner. This may provide a potential explanation for reduced *S. agalactiae* colonisation during pregnancy in women dominated by *L. crispatus*[45,84]. This has important clinical implications as *S. agalactiae* exposure from the maternal genital tract during delivery is the primary risk factor for early-onset GBS infection, which remains one of the most common infectious causes of mortality in neonates[85,86]. *L. crispatus* competition for glycan-binding sites in the cervicovaginal niche may also contribute to its capacity to dominate the niche[2,87]. Regulation of pH-dependent competition for glycan binding sites in the vaginal niche offers a strategy for the development of prebiotics designed to encourage vaginal colonisation of *L. crispatus*, to prevent pathogen colonisation and thus reducing the incidence of BV, preterm birth and associated neonatal complications[88].

In summary, this is the first study using glycan microarrays to test whole vaginal bacteria-glycan binding events with a structurally diverse collection of glycans. Our study describes a methodological approach by which customised glycan arrays can be used to identify complex bacterial binding events of biological and clinical significance. The identification of sequence-defined glycan structures specifically bound by vaginal pathogens and commensals represents an important step towards the design and development of strategies aimed to prevent or modulate bacterial binding in the vaginal niche. Although the bacteria examined in our study are associated with adverse pregnancy outcomes, these species are also more broadly implicated in other areas of reproductive tract health and disease, including bacterial vaginosis, risk of STI acquisition and HPV infection and progression of cervical cancer[89–91]. Glycomimetics, synthetic compounds mimicking native glycans, could be introduced to the cervicovaginal mucosa to block or compete for pathogen binding sites or alternatively, to encourage and support adhesion and engraftment of live biotherapeutics (probiotics). In addition, the sequence-defined glycan binding profiles of various bacteria provide valuable information for further detailed studies, such as interaction studies at the protein level or in vivo experiments. While our current study focused on the vaginal niche, the methods described can be readily adapted for investigations of colonisation and infection in other sites, including the gut, skin and respiratory tract, where collective evaluation of bacterial recognition of glycan binding can inform microbiota ecology dynamics and modulating studies.

## Methods

### Glycan materials: amino-terminating glycan probes

AEAB-terminating[92] high-mannose N-glycan probes #7-#12 were purchased from NatGlycan (Atlanta, Georgia, USA). Aminopropyl-terminating probes #6, #91-110, #170, #174-#177, and #187, synthesised chemically, were purchased from GlycoNZ (Auckland, New Zealand). The following amino-terminating glycans were obtained as gifts from different resources or collaborators: Chemically synthesised aminopropyl (Sp)-terminating probes #155 and #156 were from the Core D of the US Consortium for Functional Glycomics; probes #165 and #169 were from Prof. Nicolai Bovin (Ovchinnikov Institute of Bioorganic Chemistry, Russian Academy of Sciences), probe #182[93] from Prof Chunxia Li (Ocean University of China, Qingdao, China), and GalNAc β-linked to Ser (#163) and Thr (#164)[93] were from Ulrika Westerlind (Umeå University). The glycine (Gly)-terminating glycan probes #79-#81, #84, #151-#154, #157-159, #160-162, and #183-186 are *N*-glycosides and were prepared by conjugation of the respective reducing sugars with glycine as described[94], and purified further by HPLC. All amino-terminating glycans were analyzed by electrospray (ESI) or matrix-assisted laser desorption/ionisation (MALDI) mass spectrometry (MS) (Source Data File 1) with hexose-containing glycans quantified by micro-scale hexose assay (see below) before microarray printing.

### Preparation and analysis of amino-terminating glycan probes

**Deprotection of Fmoc-protected and per-acetylated serine- and threonine-linked mucin cores.** Probes #1-#3 and #166-#168 were prepared by deprotection of fluorenylmethoxycarbonyl (Fmoc)- and per-acetyl Ser- and Thr-linked mucin cores purchased from Sussex Research Laboratories (Ottawa, Canada). In brief, 1 N NaOCH$_3$ in methanol (~pH 10-11) was added to the solution of the protected sample (1 mg in 0.5 ml methanol). The solution was stirred for 3 h at room temperature and the pH was adjusted to pH 4-5 using acetic acid. The sample solution was dried under a N$_2$ stream and re-dissolved in 0.05 M ammonium acetate before purification by gel filtration on a Superdex Peptide column (Pharmacia, Uppsala, Sweden) with elution by 0.05 M ammonium acetate. The collected fraction was freeze-dried, and ammonium acetate removed by repeated co-evaporation with water. The deprotected samples were analyzed by MS and quantified by micro-scale hexose assay (see below) before use.

**Extended and Ser-linked mucin core 3 O-glycans.** Enzymatically prepared Ser-terminating probes #178-#180 were from Prof Kelly Moremen (Complex Carbohydrate Research Center, University of

Georgia) and were purified by gel filtration chromatography using Superdex Peptide column before use.

Ser-terminating probe #181 was chemically synthesised and as a gift from Prof Dr Akihiro Imamura of Gifu University (Yanagido, Japan). These Ser-linked and extended mucin core 3 O-glycans were analyzed by ESI-MS (Fig. 1 in Source Data File 1) and quantified by hexose assay before microarray printing.

**Preparation of amino-terminating probes from azido-terminating glycans by click chemistry.** Probes #19-#22, #27, #28, #30-39, #41, #43, #44, #48, #50, #51, #54, #56, #57, #61-#63, #65-#67, #69, #70, #72-#77 were prepared from chemoenzymatically synthesised 3-azidopropyl glycans[95] after conversion into amino-terminating glycans. Conversion of the azido- into amino-terminating glycans was performed as described previously[96] with the following modifications. To the freeze-dried mixture of azido-glycans (50 nmol) and triethylene glycol 2-aminoethyl propargyl ether (50 nmol, Sigma-Aldrich, Gillingham, England) was added pre-mixed 200 ml solution of t-butyl alcohol/water (2:1, v/v) containing $CuSO_4$ (30 nmol), sodium ascorbate (30 nmol) and *tris*-benzyltriazolylmethyl amine (TBTA, 60 nmol). The mixture was incubated at room temperature for 1 h with occasional vortexing. The solvent was removed under reduced pressure and the residue purified by gel filtration on a Superdex Peptide column as described above. Further purification was carried out by HPLC on a XBridge amide column (Waters, Wilmslow, England) with a binary solvent system of A: acetonitrile/$H_2O$ 70:30 and B: acetonitrile /$H_2O$ 20:80, with both containing 10 mM $KH_2PO_4$ (pH 3.0). The elution was carried out by gradient 5-25%B in 45 min at a flow rate of 1 ml/min and detection by UV 206 nm. The collected fractions were desalted by gel filtration on a Superdex Peptide column as described above. The converted propyl triazole-PEG4-amine (PTPA)-terminating probes were purified by HPLC (Fig. 2 in Source Data File 1) for representative HPLC profiles) and analyzed by MALDI-MS (Table 1 in Source Data File 1 for representative mass spectra). Quantitation was by micro-scale hexose assay (see below) before microarray printing.

**Preparation of AEAB-terminating probes from reducing glycans by reductive-amination.** Neutral glycan (#29, #40, #42, #45-#47, #49, #52, #53, #55, #58-#60, #64, #68, #71 and #82) and sialylated probes (#16, #17, #23-#25), purchased respectively from Eliciity (Crolles, France) or Dextra Laboratories (Reading, England), were prepared from reducing sugars whereas sulfated KS oligosaccharide probes #126-#135, were prepared from chemically synthesised reducing sugars gifted by Tokyo Chemical Industry (Tokyo, Japan). The sialylated probes #18 and #26, were prepared from the enzymatic synthesised reducing sugars (Dr Apoorva Srivastava, Department of Chemistry, Imperial College London). Preparation of the fluorescent reagent AEAB and conjugation with reducing sugars by reductive-amination were performed as previously described[92]. For purification of the conjugation product, a Sep-Pak Aminopropyl cartridge (Waters, Wilmslow, England) was used with elution by acetonitrile /$H_2O$ for the neutral sugars and acetonitrile /0.05 M ammonium acetate for the sialylated sugars. Further purification was carried out by HPLC using Amide column (XBridge, Waters). The gradient elution was acetonitrile /$H_2O$ for the neutral and acetonitrile /0.05 M ammonium acetate for the sialylated sugars at a flow rate of 1 ml/min. with detection by UV 330 nm. The detection and probe quantitation were performed using by UV 330 nm as described[92].

**Glycosaminoglycan oligosaccharide-AEAB probes.** Oligosaccharide fractions from heparin[97], chondroitin sulphate A, B and C[98], and keratan sulphate[98] were prepared by partial digestion with heparin lyase I (Sigma), chondroitinase ABC (Sigma), and keratanase II, respectively, as previously described. Hyaluronic acid (HA) fragments were prepared in two different ways. HA lyase (EC 4.2.2.1; from *Streptomyces*

*hyalurolyticus*; Sigma) was used to obtain unsaturated ΔUA-terminating oligosaccharides and HA hydrolase (EC 3.2.1.35; from bovine testes; Sigma) was used to prepare the unmodified HA oligosaccharides[99]. After conversion into AEAB probes, the conjugation products of KS DP10 and 14, (#136-#138), HA DP4, 8, 11-14 (#139-#146), and CS DP6, 10 and 14 (#111-#122) and non-sulphated chondroitin DP10, 12 and 14 (#88-#90), and the conjugation products of heparin DP6, 10 and 14 (#123-#125) were purified using an amino-cartridge (Sep-Pak Aminopropyl, Waters) with elution by sodium acetate (0.05 M). Additional purification was carried out by a short column of strong anion exchange (HiTrap Q, GE Healthcare, Amersham, England) with elution by a gradient of NaCl. The collected fractions were desalted on a Superdex Peptide column with elution by ammonium acetate (0.05 M).

**Serine- and threonine-linked sialyl mucin core 1 probes.** Mono- and disialyl core 1 probes #4, #5, #171-#173 with either Ser or Thr were isolated from human urine as described[100]. Further purification was carried out by HPLC with XBridge amide column (Waters, Wilmslow, England) with a binary solvent system of A: acetonitrile/$H_2O$ 70:30 and B: acetonitrile/$H_2O$ 20:80 (both containing 7.5 mM $KH_2PO_4$). The elution was carried out by a gradient 15-20%B in 40 min at a flow rate of 1 ml/min and detection by UV 206 nm.

**Human milk oligosaccharide AEAB probes.** AEAB probes #78 and #83-#87 were prepared from DP6-8 and DP10 fractions of HMOs as described[101]. Amide and PGC columns were used for separation and purification. Their sequences were determined by negative-ion ESI-CID-MS/MS[98].

**Complex-type N-glycan AEAB probes from fetuin N-glycans.** Fully sialylated bi- and tri-antennary N-glycan AEAB probes #13-#15, were prepared from fetuin N-glycans released by PNGase F (New England Biolabs, Ipswich, MA) and isolated from a short column of Bio-Gel P6 as described[102]. The total released N-glycans were conjugated to AEAB and the access AEAB reagent was removed by $NH_2$-cartridge. The N-glycan-AEAB fraction was fractionated by HPLC using a Hypersil APS-2 column (Fisher Scientific, Loughborough, England) with a gradient by acetonitrile/$H_2O$ containing ammonium acetate. Detection was by UV at 330 nm and the flow rate was at 1 ml/min.

**Micro-scale hexose quantitation.** The micro-scale dot orcinol assay on TLC plates was performed as previously described[103]. In brief, hexose-containing probes, especially those not AEAB conjugated, were dissolved in $H_2O$ (at approx. 0.5-1.0 mg/ml), and 1 μl was spotted onto a silica gel TLC plate. Standard solutions (Galactose in $H_2O$) were also applied alongside the glycan probes. The plate was dried and sprayed with orcinol/$H_2SO_4$ staining reagent and heated in an oven at 105 °C to develop the colour. Quantitation was carried out by scanning at 550 nm on a CAMAG TLC scanner (Omicron Research, Hungerford, England).

**MS analysis of amino-terminating glycan probes.** MS analysis of most amino-terminating probes was carried out by an electrospray mass spectrometer on a Waters (Manchester, UK) Q-TOF-type mass spectrometer SYNAPT-G2 in either positive- or negative ion mode. Cone voltage was at 50 eV or 80 eV and capillary voltage at 3 kV. Source temperature was 80 °C and the desolvation temperature 150 °C. A scan rate of 1.5 sec/scan was used and the acquired spectra were summed for presentation. For analysis, glycan probes were dissolved in $H_2O$ typically at a concentration of 10–20 pmol/μl, of which 1 μl was injected via an HPLC injector. Solvent acetonitrile/2 mM ammonium bicarbonate 1:1 was delivered by a HPLC pump (Waters) at a flow rate of 10 μl/min. Selected probes were also analyzed by MALDI-MS on an Axima MALDI Resonance mass spectrometer with a QIT-TOF

configuration (Shimadzu). A nitrogen laser was used to irradiate samples at 337 nm, with an average of 100-200 shots accumulated. A matrix solution (0.5 µl) of 2,5-dihydroxybenzoic acid (20 mg/ml) in a mixture of methanol/water (1:1) was deposited on the sample target before application of the sample solution (0.5 µl).

## Biotinylated GAG polysaccharide probes

Biotinylated CSA, CSB, CSC and heparin (#147-#150, respectively) were prepared and purified essentially as described[104]. The level of biotin introduced was one for every 50 carboxyl groups[105]. In brief, 50 mg of GAG polysaccharide were dissolved in 0.1 M MES buffer (pH 5.5) before addition of 109 ml of 25 mM biotin LC-hydrazide (Pierce, Tattenhall, UK) solution in Me$_2$SO and 130 ml of 25 mg/ml 1-ethyl-2-(2-dimethylamino propyl) carbodiimide solution in 0.1 M MES buffer. The reaction mixtures were stirred at room temperature for 24 h and dialysed extensively against deionized water. A short Sephadex G-10 column (1.6 × 30 cm) was used to remove remaining biotin and other reagents. Quantitation of GAG polysaccharides and their modified forms was carried out by carbazole assay[106].

## Bacterial culture and labelling

Bacterial isolates used in this study were commercially sourced (ATCC and DSMZ) or were kindly gifted by the Lawley and Wolfe lab (Supplementary Data 4). Isolates from the Wolfe lab were originally obtained from non-pregnant women who provided verbal and written consent following Loyola University Medical Centre Institutional Review Board approval as detailed[32]. The identity of the bacterial species used was confirmed using sanger sequencing targeting the 16S rRNA gene and MALDI_TOF.

**Bacterial Culture.** *E. coli* isolates were streaked on LB agar (Sigma, Lennox recipe) and incubated overnight at 37 °C. LB broth was inoculated from cultures on plates and incubated overnight in a shaking incubator at 37 °C and 200 rpm. Anaerobic bacteria were cultured in the following media: *L. crispatus*, MRS agar and broth media; *L. iners* and *G. vaginalis*, Columbia (Difco) + 5% v/v defibrinated horse blood (Oxoid) agar and Vaginal Microbe Media (VMM, Patent protected); *S. agalactiae*, BHI (Oxoid) agar and broth media; and *F. nucleatum*, Columbia (Difco) + 5% v/v Defibrinated horse blood (Oxoid) agar and BHI (Oxoid) broth supplemented with 5 g/l yeast extract (Sigma Aldrich), 0.005 g/l L-cysteine hydrochloride monohydrate (Merck), 0.01 g/l D-(+)-cellobiose (Sigma Aldrich) and 0.01 g/l D-(+)-maltose (0.01 g/l) media. Agar (1.2% w/v) was added to the media to prepare agar plates. Isolates of anaerobes were streaked on agar plates and incubated overnight at 37 °C in an anaerobic chamber (10% CO$_2$, 10% hydrogen, 80% nitrogen and 70% humidity). A total of 10 ml of degassed broth was inoculated and incubated overnight in anaerobic conditions at 37 °C. The starter broth was used to inoculate the required volume of fresh media and was incubated as described above until the broth reached an OD$_{600}$ ≈ 1.0.

**Fluorescent labelling of cultured live bacteria.** For staining of live bacteria, 2 ml aliquots of culture were centrifuged for 10 minutes, and the resulting pellet washed twice with 1 ml of cold sterile Hanks Balanced Saline solution (HBSS) (Gibco). 1 × 10$^8$ bacterial cells were stained with 10 µM CellTrace™ CFSE (Molecular Probes) or 10 µM CellTrace™ Far Red (Molecular Probes) in 250 uL of PBS and incubated for 1 h in a rotating wheel at room temperature protected from light. Bacteria were centrifuged, the pellet was washed twice with cold sterile HBSS and resuspended in 1 ml of cold sterile HBSS. Stained bacteria were kept on ice or at 4 °C and protected from light until further use (within 2 h for live bacteria).

**Fluorescent labelling of fixed bacterial cultures.** *E. coli* and *F. nucleatum* cultures were centrifuged for 10 minutes to pellet bacteria

and cells were washed twice in cold sterile PBS. Bacteria were fixed with 4% formaldehyde in PBS in a rotating wheel at room temperature for 15 min. *E. coli* or *F. nucleatum* fixed cells were centrifuged and washed twice in cold sterile PBS, the cell pellet was resuspended in HBSS (250 uL for 1 × 10$^8$ bacterial cells) before staining with 10 µM SYTO™59 (Invitrogen) or 10 µM SYTO™ BC Green (Invitrogen) respectively, for 30 min in a rotating wheel at room temperature protected from light. Bacteria were centrifuged, the pellet was washed twice with sterile cold PBS and resuspended in the desired volume of cold sterile PBS. Stained bacteria were kept on ice or at 4 °C and protected from light until further use (ideally no more than 1 week).

All centrifugations were done at 5000 x $g$ for 10 min in a refrigerated centrifuge set at 4 °C, except for *E. coli* where 3000 x $g$ were used to avoid fimbriae shedding. HBSS media and buffers required for staining and fixing procedures of anaerobe bacterial cultures were degassed before use. Wide-bore tips were used while handling bacterial cultures to avoid fimbriae shedding. To ensure the bacterial cultures were not contaminated during culturing, we used chromogenic agar (CHROMagar™) and light microscopy to visualise homogeneous and distinct cell morphology.

## Glycan microarray preparation and analyses

The generation of covalent sequence-defined glycan arrays and the GAG polysaccharide microarray on NHS-functionalised glass slides (Nexterion H) was performed essentially as described before[49,98]. Five oligosaccharide microarray sets and one GAG polysaccharide array set were generated (Supplementary Fig. 1 and Supplementary Data 1). Details of the preparation of the microarrays and the methods used for binding assays and data analysis are in accordance with MIRAGE (Minimum Information Required for A Glycomics Experiment) guidelines for reporting of glycan microarray-based data[107] (Supplementary Data 3).

**Binding analyses with anti-carbohydrate antibodies, plant lectins and other glycan binding proteins.** Details of the antibodies, lectins and glycan binding proteins used are summarised in Supplementary Data 2. Microarray analyses were performed at ambient temperature essentially as previously described[108]. No blocking was used for overlay assays on covalent glycan array glass slides. Pre-complexation of hSigLec15-Fc with biotinylated anti-human IgG (1:1 by weight) was performed at 4 °C for 30 min before adding it to the microarray.

**Bacterial binding analyses.** Microarray analyses of fluorescently labelled bacteria (see above) was performed as follows. Fluorescently labelled bacterial cultures at OD = 1 were resuspended in binding buffer (HBS: 10 mM HEPES at pH 7.0 or pH 4, 150 mM NaCl, supplemented with 5 mM CaCl$_2$ or Acetate buffer: 10 mM sodium acetate, at pH 7.0 or pH 4, 150 mM NaCl, supplemented with 5 mM CaCl$_2$) and were used for overlays on glycan microarrays. 100 µL of bacterial suspension was applied to the incubation chamber with the microarrays, incubated for 1 h at room temperature under mild agitation on an oscillating platform and washed three times with binding buffer, followed by two washes with HPLC grade water to remove salts from the array. Slides were dried under a mild nitrogen flow and scanned for quantitation as described below. HBS at pH4 was prepared immediately before use by adjusting the pH of HBS with HCl. Acetate buffer pH4 was prepared immediately before use by adjusting the pH with Acetic acid. The pH of the bacterial suspensions was monitored before and after overlays using pH strips.

**Microarray data analysis.** Images of fluorescently bound antibodies, lectins, glycan binding proteins and bacteria were acquired with a GenePix 4300 A scanner from Molecular Devices, quantified using GenePix software and processed with CarbArryART[109]. Bar charts and heatmaps were generated using excel. RANK quantitation is described

in MIRAGE (Supplementary Data 3) and performed as previously described.

**Glycan binding on-array inhibition assays.** For on-array inhibition studies of *F. nucleatum* 23726 glycan binding, fluorescently labelled cultures of bacteria were incubated with 10 mM D-galactose, 10 mM D-glucose, or HEPES buffer (HBS: 10 mM HEPES at pH 7.5, 150 mM NaCl, and 5 mM $CaCl_2$), for 15 min at room temperature before performing bacterial overlay on glycan microarrays.

**Bacterial competition assays on array.** For on-array competition studies between *L. crispatus* and *S. agalactiae*, 50 μl of $0.2 \times 10^8$, $0.4 \times 10^8$, $0.8 \times 10^8$, or $1.6 \times 10^8$ cells/ml of CellTrace™ CFSE labelled *L. crispatus* cultures were added to 50 μl of $0.4 \times 10^8$ cells/ml CellTrace™ Far Red fluorescently labelled cultures of *S. agalactiae* (see staining procedures described above and MIRAGE (Supplementary Data 3) before performing bacterial overlay on glycan microarrays.

## Mammalian cell culture

VK2/E6E7 vaginal epithelial cells (ATCC- CRL-2616) were cultured and maintained in Keratinocyte SFM 1X (Gibco) containing Calcium Chloride 0.4 mM and 1% w/v penicillin– streptomycin (Sigma Aldrich). Cells were cultured at 37 °C and 5% $CO_2$ in a humidified incubator and were free of mycoplasma, which was routinely tested for using the MycoStrip Detection Kit (InvivoGen).

## Chondroitinase and heparin lyase III digestion of VK2 cells

The VK2/E6E7 cells were gently scrapped and resuspended in PBS at a concentration of $8.10^5$ cells/ml. Chondroitinase ABC (E.C. 4.2.2.4, Sigma) and heparin lyase III (E.C. 4.2.2.8, recombinant, IBEX Technologies, Montreal, Canada) were added at a concentration of 0,005 U/ml and 0,001 U/ml respectively and the cells were incubated at 37 °C for 2 h. The cells were centrifuged 5 min at 300 x *g* and resuspended in PBS at the appropriate pH for antibody or bacteria binding analyses.

## Assessment of VK2-glycan and bacteria binding by flow cytometry

Sub-confluent cultures of VK2/E6E7 cells were gently scrapped and resuspended in PBS. Single cell suspensions were stained with anti-chondroitin sulphate (CS56, Abcam), anti-keratan sulphate (MZ15, a present from Fiona Watt, Imperial, London), anti-heparan sulphate (clone 10E4, Amsbio) or biotinylated hyaluronic acid binding protein (Millipore), followed by Alexa Fluor 488-conjugated anti-mIgG (ThermoFisher) or Alexa Fluor 647-conjugated anti-mIgM antibodies (ThermoFisher) or streptavidin PE (Phycoerythrin) conjugate (Miltenyibiotech), in PBS for 30 min at 4 °C. Cells were analyzed using a FACSCalibur flow cytometer (Becton Dickinson). Intact VK2 cells were gated based on their FFS and SSC, excluding cells with extreme low or high FSC and SSC values, see example in Supplementary Fig. 8). Flow cytometry data was analyzed using CellQuest software (BD Biosciences).

For bacterial binding assays, VK2/E6E7 cells were added to bacteria suspensions to obtain a ratio of bacterial particles to cells or MOI (Multiplicity of Infection)=20 for all bacterial strains except for *G. vaginalis* at pH 4 where MOI = 100 was used. Cell mixtures were gently stirred for 1 h at 4 °C. Where appropriate, fluorescently labelled bacteria were incubated in PBS containing polysaccharides at 5 mg/ml for 30 min before binding assays. Cells were analyzed using a FACSCalibur flow cytometer (Becton Dickinson) and analyzed using the CellQuest software (BD Biosciences).

## Electron microscopy

Overnight cultures of *E. coli* isolates were imaged by negative stain. For staining, 10 μl of culture was deposited onto glow-discharged 300 mesh carbon-coated copper grids (Agar Scientific), and left for 2 min.

The grid was washed twice with 10 μl of water. The sample was stained with 0.2% (w/v) or 0.04% (w/v) phosphotungstic acid for 7 sec, after which the stain was blotted off. The grid was imaged on a Tecnai 12 Spirit transmission electron microscope (FEI) operating at 120 kV and with a 2 K eagle camera (FEI).

## SaHyal_767 protein expression and purification

The gene WP_000403395.1 encoding a putative hyaluronate lyase enzyme from *S. agalactiae* 767 was analyzed using InterProScan. The designed construct *Sa_Hyal_767* includes both the carbohydrate-binding and catalytic domain (residues 106-1081). Tyr578 was mutated to Phe to reduce enzymatic activity on glycan binding assays and was codon optimised for *E.coli* expression. The construct was synthesised and cloned between the BamHI/XhoI sites in the multiple cloning site of the expression vector pET28a(+) (GenScript) (Supplementary Fig. 10). The resulting recombinant protein has a hexa-histidine-tag at the N-terminal. Protein expression was carried out in *E. coli* strain BL21(DE3) induced by adding 1 mM isopropyl-β-d-thiogalactopyranoside, overnight at 19 °C. IMAC purification was performed with a imidazole gradient. Fractions containing the purified recombinant proteins were buffer exchanged to 50 mM HEPES, pH 7.5, 100 mM NaCl, 5 mM $CaCl_2$ and 0.02% sodium azide. The final yield of purified protein was approximately 8 mg/L of culture.

## Isothermal calorimetry (ITC) assays

The theoretical basis of kinetic measurements by ITC is described elsewhere[110]. Briefly, in a single injection ITC experiment, the total heat measured is proportional to the apparent enthalpy (ΔHapp), and the number of moles of product generated. The reaction rate can be related to the amount of heat generated over time. The affinity for the substrate (Km) and the turnover rate (kcat) values of the enzyme can be obtained by fitting the rate of the reaction versus the substrate concentration plots with a Michaelis-Menten curve. Experiments were performed on MicroCal PEAQ-ITC calorimeter (Malvern). One single 38 μl injection was performed with a speed of 0.5 μL $s^{-1}$. The instrument was set to high-feedback mode, reference power 5 μcal $sec^{-1}$, stirring speed of 650 rpm, and an experimental temperature of 25 °C. A 600 second pre-injection delay was applied for baseline stabilisation after equilibration. Apparent enthalpy of the reaction, heat rate (dQ/dt), and Michaelis–Menten plots were generated using MicroCal PEAQ-ITC Analysis Software (Malvern).

*Sa_Hyal_767* activity against HA 14-mer, heparin 14-mer, and LNT was tested at pH 7.5. The enzyme was dialysed overnight into a buffer either containing 50 mM HEPES at pH 7.5, 150 mM NaCl, and 5 mM $CaCl_2$. The lyophilised glycans were dissolved in the same corresponding buffer. A *Sa_Hyal_767* solution at 50 nM was placed in the sample cell, and a 0.5 mM glycan solution was loaded in the injection syringe. Substrate autohydrolysis and heat of dilution phenomena were assessed by replacing the enzyme solution in the calorimetric cell with reaction buffer and carrying out the measurements with identical experimental settings.

## Statistical analysis

In glycan microarray binding analyses each assay was performed with four technical replicates and results in graphs presented as mean +/− SD of quadruplicates (Source Data Files 2–7). Bacterial binding to polysaccharide GAG arrays was performed in three independent assays. Results in graphs are presented as mean +/− SD of quadruplicates on the microarray of representative experiments or as mean +/− SD of three independent experiments. Statistical significance of data from independent experiments was assessed using an unpaired two-tailed Student's t test. Microscopy images are representative and similar results were observed in different fields of view with two independent biological replicates. Mammalian cell-based assays were performed with at least three biological replicates. Results in graphs are

**Article**

presented as mean +/− SD. Statistical significance was assessed using 2-way ANOVA.

### Reporting summary

Further information on research design is available in the Nature Portfolio Reporting Summary linked to this article.

## Data availability

Source data and glycan microarray data are provided with this paper as Source Data Files. Original glycan microarray image files and gpr files are available upon request. Glycan microarray processed data is provided in Source Data Files 2-6. Glycan microarray metadata is summarised in Supplementary Data 3 with MIRAGE information. There are no restrictions on the data availability. Source data are provided with this paper.

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

## Acknowledgements

This work has been funded by the March of Dimes European Prematurity Research Centre grant 22-FY18-82 awarded to Imperial College London (ICL). The glycan microarray studies were performed in the ICL Carbohydrate Microarray Facility supported by the Wellcome Trust Biomedical Resource Grants (WT099197/Z/12/Z, 108430/Z/15/Z and 218304/Z/19/Z). The sequence-defined glycan microarrays contain oligosaccharides provided by collaborators including CFG Core D, Nicolai Bovin, Chunxia Li, Akihiro Imamura, Ulrika Westerlind, Kelly Moremen, and Apoorva Srivastava whom we thank. J.R.M. and the Division of Digestive Diseases at ICL receive financial and infrastructure support from the NIHR Imperial Biomedical Research Centre (BRC) based at ICL and ICL Healthcare NHS Trust. L.S. and B.G.M. are supported by Imperial Health Charity in partnership with the Parasol Foundation and the Rosetrees Trust. P.R.B. and L.S. receive infrastructure support from the NIHR Imperial Biomedical Research Centre (BRC) based at ICL and IC Healthcare NHS Trust. Y.L., A.S.P. and B.A.P., thank the GLYCOTwinning project HORIZON-WIDERA-2021-101079417 funded by European the Commission and a DL-57/2016 Program Contract to B.A.P. We would like to acknowledge the Centre for Biomolecular Spectroscopy at King's College London that is funded by the Wellcome Trust and the British Heart Foundation (WT202767/Z/16/Z and IG/16/2/32273). V.P. receives financial support of the Ministry of Science and Higher Education of the Russian Federation (Contract No. 075-00277-24-00). The authors thank Prof Barbara Mulloy for her help in NMR data interpretation and Dr. Daniel Slade for the delta *galkt* and delta *galkt fap2 F. nucleatum* 23726 mutants.

## Author contributions

V.T.O., Y.L. and D.A.M. designed the work and wrote the manuscript with input from all authors. V.T.O. performed the bacterial microarray assays, the microarray quality control with help from L. R. and analysed the data.

W.C. and Y.Z. prepared the glycan probes, A.D.M. printed the glycan microarrays and with V.T.O. established the optimal conditions for whole bacteria binding used in this manuscript. V.T.O. and B.G.M. set up and adapted the bacterial staining protocols. L.A.R. prepared the bacterial samples under J.R.M. supervision. A.C.D. designed and performed the flow cytometry based bacterial binding to VEC assays. A.S.P. and B.A.P. produced the full-length hyaluronidase and G.D.N performed the ITC assays. B.G.M. and Y.S.L. expanded the initial bacterial collection received from the Wolfe and Lawley labs. H.C. and V.P. produced some of the glycans used for this work. B.G.M. and H.A. performed and analysed the electron microscopy with help from T.R.D.C. Y.A. helped with annotation of glycan probes on *in house* databases. L.S., P.R.B., J.R.M. and T.F. provided useful insight and valuable input for the scientific work and experimental design.

## Competing interests

The authors declare no competing interests.

## Additional information

[1]March of Dimes Prematurity Research Centre at Imperial College London, London, UK. [2]Glycosciences Laboratory, Department of Metabolism, Digestion and Reproduction, Imperial College London, London, UK. [3]Institute of Reproductive and Developmental Biology, Department of Metabolism, Digestion and Reproduction, Imperial College London, London, UK. [4]Division of Digestive Diseases, Department of Metabolism Digestion and Reproduction, Imperial College London, London, UK. [5]Division of Biomedical Sciences, Warwick Medical School, University of Warwick, Coventry CV2 2DX, UK. [6]UCIBIO, Applied Molecular Biosciences Unit, Department of Chemistry, and Associate Laboratory i4HB-Institute for Health and Bioeconomy, School of Science and Technology, NOVA University Lisbon, Caparica, Portugal. [7]Randall Division of Biophysics, New Hunt's House, Guy's Campus, King's College London, London, UK. [8]The Parasol Foundation for Women's Health and Cancer Research, Imperial College Healthcare NHS Trust, London, UK. [9]School of Medicine and Pharmacy, Ocean University of China, Qingdao 266237, China. [10]Nesmeyanov Institute of Organoelement Compounds, Russian Academy of Sciences, Moscow, Russia. [11]Centre for Bacterial Resistance Biology, Department of Life Sciences, Imperial College London, London, UK. [12]Robinson Research Institute, University of Adelaide, Adelaide, Australia.

