## [Transparent Peer Review file · Nature Communications]

Identification and characterisation of vaginal bacteria-glycan interactions implicated in reproductive tract health and pregnancy outcomes

Corresponding Author: Dr Virginia Tajadura Ortega

Version 0:

Reviewer comments:

Reviewer #1

(Remarks to the Author)

The vaginal microbiome plays an important role in human health and reproduction. The mechanistic factors that contribute to establishing and maintaining the microbiome, as well as problems that cause imbalances, are important to understand for both prevention and treatment of associated diseases. In this study, the authors construct a series of glycan microarrays and use them to profile carbohydrate-binding properties of various vaginal bacteria. Overall, the authors evaluated binding of 22 strains of bacteria on glycan microarrays. The resulting information provides a valuable resource for studying the molecular basis for adhesion and retention of commensal vs pathogenic microorganisms. The paper is well-written, the data are solid, and there is a wealth of information included in the study.

One concern is that the importance of the work and the connections between their binding profiles and human health may not be clear to all readers. It would help if the authors described the connections in more detail and clarified the importance of the results. For example, the authors mention that the information might be used to identify glycan targets for further studies – this statement is a little vague. What sort of further studies? How important is the new information to enabling those studies? Perhaps they could explain a little better. They mention that their results open the way to broader studies on blood group expression and bacterial binding in clinical cohorts. This sound potentially very important but a bit vague – perhaps they could explain a little better. What new studies were opened? As another example, they say their findings support the use of *L. crispatus* as a live biotherapeutic for preventing preterm birth and complications. Again, potentially interesting but a bit vague – how exactly do their results support this clinical avenue?

At times, it can be difficult to determine which results/information are new and which results are expected based on prior studies. For example, for the 22 strains, were these all known previously to bind carbohydrates? If so, was there any specificity information previously published? If not, which ones were not previously known to bind carbohydrates? The amount of new information significantly affects the potential impact of this study.

Certain parts of the Results section were a little confusing:

-In the first section, the authors state that they constructed an array with 187 glycans and used it to profile 6 bacteria. Then they stated the evaluated binding of *E coli* 600 on a 69 glycan subarray. This seems a little confusing – what happened to the 187 glycan array, and why didn't they use that? Then they talk about profiling other bacteria in this section, but it is not clear if this is with the 187 glycans or the 69 glycan subarray. Please clarify.

-The also talk about profiling ConA, but it is not clear if this is with the 187 array or 69 subarray. Please clarify

-Next, they talk about profiling 22 strains with an array containing 110 neutral and sialylated probes. Is this a subset of the 187 glycans? If so, why didn't they profile binding towards the full 187? Please clarify. If it's a different set, why not also use the 187 glycans to complement these 110?

-in the first section, the text says that blood group antigens were not bound, but in a later section they talk about blood group antigen binding. This is a bit confusing. They hint that it might be a linker effect, but this should be clarified in the text.

-It sounds like they are using several different arrays – 187, 110, GAGs. Readers might wonder why they are using separate arrays rather than one big array? Wouldn't that save a lot of time and facilitate comparisons?

On page 8 line 187, the text says that binding to AEAB glycan probe linker was strongest on covalent arrays suggesting presentation is important. The wording here is a little confusing, as it implies that screening was also done on non-covalent arrays or some other array format. I didn't see any mention of non-covalent arrays. Were non-covalent arrays used? If so, what were the results? Some clarification would help here.

In the second paragraph of the Discussion, they talk about setting up conditions for screening bacteria. Was this straightforward? Were there any problems getting good binding and removing excess cells? Were there any special conditions or procedures to get good data? If it was fairly easy, no need to discuss. If there were problems, it would be nice to mention them and how they solved them – in the Results, Methods, or Discussion.

Methods

The authors mention profiling the arrays with a bunch of antibodies and lectins in the experimental section. However, there is no mention of this in the main text. What were the results? Were they as expected? For example, their anti-LeY antibody bound well to LNnFPI-AEAB, H Tetra T2-Lac-PTPA, and Hexa T1-T2-Lac-PTPA – was that expected? There are some pretty big differences between the two BG-A antibodies, T36 and Z2A. was that expected? The LeA antibody binds sialyl-LeA and sulfo-LeA – is that expected? Also, how do the results for the lectins and antibodies help with interpreting results. For example, they talk about bacteria not binding blood group antigens (page 6) – were these bound by the anti-blood group antibodies? This information would confirm that it's a real effect and not a printing issue.

Characterization data. The authors state that many of the compounds were characterized by MS. However, a number of others don't mention any characterization data, such as AEAB, Ser/Thr core 1 probes. The authors should provide at least some characterization data for all new compounds described in this paper. Ideally, this would include MS and NMR. however, I realize they may have only gotten small amounts of many of them, making NMR difficult. Where possible, some data for composition and purity should be included for the compounds. Additionally, they should include copies of the actual data (e.g. MS spectra) in the supporting information. In cases where this is not feasible, they should acknowledge that the purity and/or composition has not been determined (perhaps in Extended Table 1).

I had some difficulty figuring out how many bacteria were incubated on the array and what volume they were in. From what I can tell, they bacteria were grown to an OD of 1.0 and then pelleted. What volume of culture was pelleted (or how many total cells were pelleted)? What volume were they resuspended in?

For the SaHyal_767 protein expression, the authors need to include the exact nucleotide sequence used for the expression. For example, was it codon optimized? They should also include the final yield from a stated volume, so that others would know how much to expect if they repeated this work.

In the Fluorescent labeling section, please add the volume of PBS used.

In the labeling of fixed bacteria, please add the volume of HBSS used to resuspend the cells before staining.

Reviewer #2

(Remarks to the Author)

This study, "Identification and Characterization of vaginal bacteria-glycan interactions implicated in reproductive tract health and disease" is interesting and important. The use of glycan arrays allows a comprehensive approach to identify the glycan binding partners of several bacterial species that inhabit this space. Following up the glycan arrays with binding assays using in vitro-grown vaginal epithelial cells is also a strength. Although it is only applied to one glycan and one specific bacterium, some of the flow cytometry experiments using enzymes and competing glycans are elegant in design and their results point towards the same conclusion. Together the data support the authors' hypothesis that chondroitin sulfate (CSC) is a ligand for *Lactobacillus crispatus* binding to epithelial cells. This is significant. Unfortunately, there appears to be a major flaw in the experimental design for the binding assays. The authors have been using both PBS and HEPES to buffer at pH 7 and pH 4. With multiple ionizable groups, these pH ranges work well for PBS. However, HEPES, which is used in the binding buffer bacteria are resuspended in, does NOT have sufficient buffering capacity at pH 4 to make the conclusions about pH that the manuscript is putting forward (pH dependence is concluded throughout based on these methods).

MAJOR COMMENTS:

The focus on pH in the vagina is relevant. The methods used can have great impact on how the findings are interpreted. Please provide more information about how buffers were prepared and used.

PBS: There are many ways to prepare PBS. The authors here use "PBS" at two different pHs (4 and 7) to make conclusions about "pH-dependent" binding throughout the paper. It is of key importance to clearly describe these methods. How was the buffer prepared and pH-titrated? PBS has two pH ranges where it can act as an effective buffer. Did the authors ensure

that buffering capacity was maintained during the binding assays? There are wash and resuspension steps should be better described too.

HEPES: It is very concerning that the authors seem to be stating (line 618) that they are using HEPES “at pH 7.5 or pH 4” as the “binding buffer”. HEPES is not an effective buffer at pH 4. If the majority of the studies were carried out using this strategy, it seems to call into question whether pH 4 is accurate. I do not disbelieve that the studies show a pH dependence, but the facts do matter. It does not have to be pH 4 to place into biological context (see next comment), but this error must be corrected!

The pH ranges associated with the *Lactobacillus*-dominant microbiotas is around pH 4.0-4.4 for *Lactobacillus*-dominant microbiotas (CSTI and CSTIII) and pH 5.3-5.5 for BV/CST IV microbiotas (Ravel, 2011, PNAS). pH 4 is relevant, but using pH 7 limits the conclusions and biological interpretations one can make about how these glycan-binding interactions might operate in the vagina these major community state types. Though not physiologically impossible, pH 7 would be an outlier in BV, where the average is ~ pH 5.5. In light of the issues raised above about pH and buffering capacity, it would be good to take another stab at this.

Figure 1A needs to be a lot bigger – this is a key figure that enables the reader to see the scope of your work!. The rest of this figure is not necessary to show in the main body of the paper. This validation of the array can go to the extended figures. It would be better to provide examples of the classes of glycans tested on the array (like Figure 3 SNFG glycans; this is very nice!)

I find Extended Figure 2 to be much more informative than Figure 2 and feel that these figures should be switched (Extended Figure 2 should be shown along with the main text). The highlighted boxes labeled with the different types of glycans that is shown in current Fig 2 is nice. Please put this in both figures.

Binding assays are at ambient temperature and with live bacteria. How can you be sure the bacteria are binding to the glycan presented in the array? In the presence of bacterial exoglycosidase activities, it could be reasoned that some bacteria remove outer monosaccharides to bind inner glycan epitopes. For example, the authors find that virtually all of the bacteria bind to sialylated glycans in multiple presentations. Some of these bacteria (e.g. *Gardnerella*) have also been found to express glycosidase activities and might bind underlying glycan structures. This should be considered and the arguments presented for how the array technology might be able to tease this apart.

It is not clear that the data support the vague interpretation in the abstract and elsewhere that glycan binding signatures of opportunistic pathogens were “highly distinct” from commensals.

First, regarding the definition of “commensals”: In the abstract, the authors refer to *Lactobacillus iners* and *Gardnerella* as “commensals”, but by definition commensals do not cause harm. There are a number of publications using tissue and animal models showing that both microbes express cytolytic toxins (ineroysin, vaginolysin) capable of lysing human cells. This seems like a strong argument against the idea that these microbes exist peacefully without causing harm to their host. This is further supported by health problems that have been associated with the community state types where these taxa are dominant (in comparison to *L. crispatus*).

If the authors plan to say glycan binding was “highly distinct” from one group of bacteria to another, please adequately justify the categorization of bacteria and also summarize the findings that support such a statement or make the statement more specific.

Likewise at the end of the Introduction, the authors conclude that “our results reveal specific and distinctive glycan binding profiles of vaginal commensals and pathogens”. It is not clear what specific and distinctive results support this statement. Except for the couple of examples already known (*E. coli* FimH and mannose, *F. nucleatum* Fap2), it seems like there is actually more in common across the microbes than there are differences. Please make a concluding statement that is more specific to the findings.

Context and Physiological Relevance of bacterial-glycan interactions:

Figure 1 (*E. coli*) and Figure 3 (*F. nucleatum*) are relevant to validate the arrays because of known binding activities between these bacteria and relevant glycans. These are nice figures. However, it is surprising given the availability of FimH mutants in UPEC and Fap2 mutants in *F. nucleatum* that the authors did not obtain and use these mutants to further establish the specificity of binding both in terms of the glycans and the lectins that recognize them. There is speculation in the discussion that additional binding activities seen for *F. nucleatum* is “possibly mediated by Fap2 or adhesins yet to be identified.” This is testable and would provide a substantial advance by revealing Fap-2-independent processes not yet discovered.

The authors conclude (line 266) that the EM data shown for *E. coli* “provided a structural explanation for the binding to high mannose structures.” This statement seems disingenuous, given the large literature describing interactions of UPEC FimH-adhesin that binds mannose structures. The “structural explanation” claimed by the authors pales in comparison to what is known already in the literature.

There is a single weighty sentence spanning lines 186 to 180. However, it seems to be missing information and is not understandable. The sentence states bacterial binding was strongest “on covalent arrays” but in comparison to what? It is not clear why the authors conclude that this suggests “that presentation of glycans within the chemical environment is an important mediator of interactions.” Are the authors referring to background binding to the chemical linkers used for some of the glycans? If so, this needs to be stated more clearly.

Heparin did not get a proper introduction in the manuscript and is presented as a finding without any context. It is not clear

why heparin would be relevant to the vaginal mucosa.

Re: "Nonsulfated" chondroitin sulfate doesn't seem relevant to lectin-glycan interactions between host and vaginal bacteria – aren't these precursors inside cells that are used to build CS?

The final sentence of the abstract "This study highlights glycans as key mediators of vaginal bacterial binding events and targets for microbiome modulating therapeutic strategies," is overly broad and needs to be qualified better. There is much to do to ascertain the key binding events for vaginal bacterial binding and how it participates in colonization or pathogenesis. I do NOT think this study succeeds in highlighting "glycans as targets for microbiome modulating strategies". This is a huge leap.

Can the authors clarify exactly where they used fixed bacteria prior to some of the binding assays? Is it only Figure 1 and extended Figure 1 that used fixed bacteria? If not and this strategy is sprinkled throughout the paper, can the authors please concisely state in the figure legends whether bacteria are fixed and labeled prior to binding or whether they undergo the live labeling strategy?

The data presented in Figure 5B seems problematic. The histogram on the left shows HUGE binding at pH7 by *L. crispatus* with a peak intensity around 10^3 but the right side shows data in which the signal seems to be maxing out a little over 10^2 .

Extended Figure 6. It seems problematic that the methods for panel C (*G. vaginalis* 775) compare the pH 4 and pH 7 conditions using different MOIs. How do the authors justify that they used MOI of 100 only for the *G. vaginalis* pH 7 condition?

The discussion refers to prior work from this group (ref 41 line 277) that made computational predictions of possible lectins from vaginal bacteria. It seems the current dataset does not bear out those predictions (?). I would welcome a paragraph about this in the discussion.

MINOR:

First sentence of the abstract uses causal language, "increases risk" should be "is associated with higher risk of"

HEPES buffer should be in all caps, line 618 and possibly other places, please find and replace

Line 575; please specify a maximum time that labeled bacteria were stored as stated prior to use.

Extended figure 5 Title should say "bound by *F. nucleatum*" not "bound by bacterial of the vaginal microbiome", since only *F. nucleatum* data is presented in the figure.

There are two panel (B)s listed in the legend for Figure four, the second one should be (C).

Line 240– should refer to Fig 5B not 6B

Line 246 should refer to Fig 5C not 6C

Please remove "recently" from line 320 and provide an earlier reference. This was demonstrated a long time ago.

Line 610. The methods mention using Siglec15-Fc reagents, but the manuscript does not seem to include this.

The statement line 241-242, "*S. agalactiae* and *L. crispatus* rarely coexist in the vaginal niche" is incorrect and not supported by the paper cited, which does not name taxa to the species level.

Reviewer #3

(Remarks to the Author)

This is very nice descriptive work detailing glycan interactions with some key vaginal bacteria, GBS and pathobionts. The significance of the work is centered on preterm birth, PPRM and neonatal outcomes. As such, the title should be revised to reflect the focus of the paper and the discussion. In terms of the clinical relevance and translation of the work, the authors suggest in the last sentence of the discussion that *L. crispatus* based live bio therapeutics for preventing preterm birth, but this is not a new or original finding resulting from this work. It is however another aspect of *L. crispatus* dominant microbiomes that could suggest a potential rationale for outcompeting pathogenic bacteria. Unfortunately, the authors do not demonstrate that phenomenon in the work. Additional evidence is needed.

Abstract/Intro: It is not clear why the specific bacterial genus/species were selected for the study? For example, *Prevotella* is a key vaginal bacteria and known to interact with glycans...why wasn't that bacteria studied? There is no description or rationale outlined for the specific bacteria and species that were included. There has been a restructuring of *G. vaginalis* and other key species identified..how does that relate to the work presented herein? Why did the authors select the *G. vaginalis* that was included in the study? Suggest adding a rationale for the selection of these bacteria, species, # of strains tested as strain specific differences exist.

Results: The data analysis and interpretation are clear and methodology is sound for the glycan analysis. More glycan interaction work elucidating a mechanism is needed. It is not clear what is gained by the in vitro experiments. Do the authors anticipate that culturing E6/E7 immortalized cells in monolayer would recapitulate the in vivo representation of glycan presentation? This was not acknowledged in the manuscript. Mucin expression is impacted by polarization of epithelial cells and monolayers are not polarized and do not have the same levels of expression. How does this relate to glycan expression?

The methods are detailed to allow for reproduction.

Discussion: The discussion could be further enhanced with more of a clinical/translational perspective. Again it is focused primarily on obstetric health and more specifically preterm birth so the title should reflect that focus. What are the significance of the findings? Is this data providing more of a compelling argument for *L. crispatus* role in full term pregnancies? This was

not clear beyond the work that has been published on blood groups previously and a suggestion towards *L.crispatus* based biotherapeutics for microbiome modulation. The work is solid, but overall descriptive and I was left wanting more of the mechanism, clinical significance and impact on the field in this section. What is the future application?

Version 1:

Reviewer comments:

Reviewer #1

(Remarks to the Author)

the authors have addressed all of my concerns

Reviewer #2

(Remarks to the Author)

The authors have sufficiently revised the manuscript to address my concerns.

Authors Response

Reviewer #1

We appreciate Reviewer #1's overall positive assessment of our manuscript that "The resulting information provides a valuable resource for studying the molecular basis for adhesion and retention of commensal vs pathogenic microorganisms. The paper is well-written, the data are solid, and there is a wealth of information included in the study."

This reviewer expressed a concern "that the importance of the work and the connections between their binding profiles and human health may not be clear to all readers. It would help if the authors described the connections in more detail and clarified the importance of the results."

Response: *We thank the reviewer for their constructive comments and appreciate the opportunity to clarify and explain the importance of our results in more detail. As highlighted by the reviewer, our study describes a methodological approach by which customised glycan arrays can be used to identify important bacterial binding events of biological and clinical significance. While our current study focused on investigating binding events in the vaginal niche relevant to reproductive health, there are many other areas where the identification of glycan targets would provide molecular insight into key adhesion events involved in colonisation and infection, including the respiratory tract, gut and skin niches.*

The identification of glycan structures bound specifically by vaginal pathogens and commensals also represents a critical step for the design and development of strategies for preventing or modulating bacterial binding in the vaginal niche. For example, it is feasible that glycomimetics, naturally derived or synthetic chemical entities designed to mimic the structure of native glycans, could be introduced into the cervicovaginal mucosal interface to compete for pathogen binding sites. Alternatively, the glycans bound competitively by commensal species could also be introduced to encourage and support adhesion and engraftment of live biotherapeutics (probiotics).

We have now modified the discussion of our article to provide better connections between the results presented and our understanding of human health and disease.

1.2 They mention that their results open the way to broader studies on blood group expression and bacterial binding in clinical cohorts. This sound potentially very important but a bit vague – perhaps they could explain a little better. What new studies were opened?

Response: *Histo-blood group (ABO) antigens are presented on Type-1 and Type-2 antennae of O- and N-glycans, and Type-3 backbones on O-glycans. The expression of these carbohydrate sequences is regulated by the activity of FUT1 and ABO genes, which encode fucosyltransferases and glycosyltransferases, respectively. Additional glycan modification on epithelial cells and mucosal secretions are determined by the FUT2 gene, which encodes the $\alpha(1, 2)$ -fucosyltransferase enzyme responsible for the addition of a fucose residue to the terminal galactose on a type 1 glycan precursor forming the H antigen. Up to 20% of human populations carry mutations in the FUT2 gene that result in non-secretion of these antigens from mucosal surfaces.*

Previous studies have shown that ABO blood group phenotype and secretor status associates with susceptibility to pathogen-driven disease, since the ABO(H) antigens can act as adhesion molecules and a source of nutrients for bacteria. For example, blood group B individuals are more susceptible to Group B Streptococcus infections while blood group O individuals are more vulnerable to H. pylori infection and development of peptic ulcer. This opens the possibility of examining blood group antigen-microbiota binding to stratify infectious disease risk in different clinical contexts.

Our microarray-based approach provides an effective screening tool for characterising interactions between bacteria and blood group glycans. We show that vaginal pathogens and commensals have the ability to bind several blood group glycans, present in CVF N- and O-glycans^{1, 2}. In this way our findings may add mechanistic insight into recent reports of increased preterm birth risk in non-secretor individuals harbouring pathogen enriched vaginal microbiomes³. Further studies investigating the relationship of microbiota composition, HBGA expression and blood group antigen binding and utilization by bacteria in pathogen-driven disease, particularly in the context of reproductive tract infections, are warranted. We have now included these points in the discussion.

1.3 As another example, they say their findings support the use of *L. crispatus* as a live biotherapeutic for preventing preterm birth and complications. Again, potentially interesting but a bit vague – how exactly do their results support this clinical avenue?.

Response: *There is substantial associative evidence that *L. crispatus* provides protection against preterm birth and other obstetric complications, however the mechanisms by which it achieves this are not fully elucidated. Our findings show that *L. crispatus* competes for binding to specific glycan structures that are also targeted by vaginal pathogens, such as Group B Streptococcus (*S. agalactiae*), which is known to be associated with increased preterm birth risk and neonatal sepsis.*

*We also present evidence that vaginal pathogen-glycan binding is largely pH dependent. These results, taken together with the knowledge that *L. crispatus* colonisation of the vagina is associated with an acidic pH (Mirmonsef, P. et al. PloS One, 2014)⁴, provide a mechanistic basis for using *L. crispatus*-based live biotherapeutics as a therapeutic*

strategy to prevent pathogen colonisation in pregnancy and therefore reduce risk of adverse outcomes such as preterm birth.

We have substantially modified the text to include this information (lines 410-422).

1.4 At times, it can be difficult to determine which results/information are new and which results are expected based on prior studies. For example, for the 22 strains, were these all known previously to bind carbohydrates? If so, was there any specificity information previously published? If not, which ones were not previously known to bind carbohydrates? The amount of new information significantly affects the potential impact of this study.

Response: *The work presented in our article is almost entirely novel. This is the first description of using glycan microarrays to test whole bacterial strain-glycan binding events with such a complex and comprehensive collection of glycan structures. As described, a small number of the binding events we observed had been previously reported in other contexts using alternative methods. These results provide sound validation for our methodological approach.*

For example, the strain E. coli C600 is known to bind to oligomannose N-glycans. We used this knowledge to establish and validate binding conditions on our glycan microarrays. Some knowledge exists on the recognition of Gal-terminating glycans by F. nucleatum. However, our study is the first to provide direct evidence at the glycan sequence level. The remaining observed binding activities for the other vaginal bacterial strains examined were almost all entirely novel.

*We have updated the text to ensure clarity around reporting of new findings, and already described glycan binding events (see lines 119, 171, 196, 220, 357). A succinct summary of results has also been added at the end of the MIRAGE document in **Supplementary File 3**.*

1.5 Certain parts of the Results section were a little confusing: -In the first section, the authors state that they constructed an array with 187 glycans and used it to profile 6 bacteria. Then they stated the evaluated binding of E coli 600 on a 69 glycan subarray. This seems a little confusing – what happened to the 187 glycan array, and why didn't they use that? Then they talk about profiling other bacteria in this section, but it is not clear if this is with the 187 glycans or the 69 glycan subarray. Please clarify.

Response: *We thank the reviewer for highlighting this. We have now updated the manuscript throughout to communicate more clearly which arrays were used in each experiment.*

Briefly, the 187 glycan probes used in the study are described in **Extended Table 1**. These were included in six different glycan Microarray sets as detailed in new **Extended Figure 1** and **Supplementary File 3** (MIRAGE document). The specific glycan probes included in each subarray along with quality control array data using glycan binding proteins with known specificities are presented in **Supplementary File 2**.

Microarray set 1, which consisted of 69 probes, was used initially to test binding conditions of four *E. coli* strains as well as binding of live and fixed cultures of *E. coli* C600.

Additional glycan binding experiments for the full bacterial collection (22 different strains) was performed using a set of 4 microarrays totalling 146 different glycan probes as follows:

Microarray set 2a and set 2b, which consisted of 90 probes covering neutral and sialylated glycans, and including 32 of the probes originally included on Microarray set 1, all printed at 300 mM. *Due to limited amounts of some of the glycans probes included in Microarray set 1, only a subset of those were included in subsequent Microarray set 2 for testing with the whole collection of bacteria.

Microarray set 3, which consisted of 59 probes printed at 100 mM and included 39 glycan probes with a PTPA linker (also included in Microarray set 2b) and 20 glycan probes with a sp linker. It was also used to compare binding of live and fixed cultures of *F. nucleatum*.

Microarray set 4, which consisted of 36 GAG oligosaccharide probes.

Microarray set 5, which consisted of 4 GAG polysaccharide probes was used with selected bacteria only.

We have now clarified this information throughout the text and included a visual representation of the array sets used in new **Extended Figure 1**.

1.6 The also talk about profiling ConA, but it is not clear if this is with the 187 array or 69 subarray. Please clarify.

Response: ConA was used as a control (line 128) for the oligo-mannose binding on the *E. coli* binding assay on Microarray set 1, data in **Extended Figure 2A**. It has also been used for the quality control of the other subarrays when oligomannose glycans are present (**Supplementary File 2**).

1.7 Next, they talk about profiling 22 strains with an array containing 110 neutral and sialylated probes. Is this a subset of the 187 glycans? If so, why didn't they profile

binding towards the full 187? Please clarify. If it's a different set, why not also use the 187 glycans to complement these 110?

Response: *Please see above response to point 1.5. We have now amended the text to clarify.*

1.8 In the first section, the text says that blood group antigens were not bound, but in a later section they talk about blood group antigen binding. This is a bit confusing. They hint that it might be a linker effect, but this should be clarified in the text.

Response: *We understand the reviewer is referring here to the description where we refer to the absence of binding to short glycan chains with a sp linker. We observed binding of bacteria to blood group epitopes on longer glycan chains with an AEAB or PTPA linker. While it is possible that presentation mediated by different linkers affects binding, in this case, the short length of the glycan chains might also impair binding.*

*The linker effect mentioned previously in the Results section refers to a broader observation, mainly by Gram positive bacteria, which exclusively were binding glycans immobilised through an AEAB linker. However, binding to blood group glycans was weak and only reproducibly bound by bacteria when printed at high concentration/density (300 mM). We have rephrased for clarity and added a paragraph in the discussion (lines 307-319) and in the MIRAGE document (**Supplementary File 3**).*

1.9 It sounds like they are using several different arrays – 187, 110, GAGs. Readers might wonder why they are using separate arrays rather than one big array? Wouldn't that save a lot of time and facilitate comparisons?

Response: *Please see above response to point 1.5. We have now specified and clarified the use and purpose of several microarray sets throughout the text and included a visual representation of the array sets used in new **Extended Figure 1**. Some of the microarray sets were used only for comparisons of live and fixed bacteria (ie: Microarray set 1) or for validation of our results (ie: Microarray set 5).*

1.10 On page 8 line 187, the text says that binding to AEAB glycan probe linker was strongest on covalent arrays suggesting presentation is important. The wording here is a little confusing, as it implies that screening was also done on non-covalent arrays or some other array format. I didn't see any mention of non-covalent arrays. Were non-covalent arrays used? If so, what were the results? Some clarification would help here.

Response: We thank the reviewer for pointing this out. Only covalent arrays were used in this study. Please see above response to point 1.8.

1.11 In the second paragraph of the Discussion, they talk about setting up conditions for screening bacteria. Was this straightforward? Were there any problems getting good binding and removing excess cells? Were there any special conditions or procedures to get good data? If it was fairly easy, no need to discuss. If there were problems, it would be nice to mention them and how they solved them – in the Results, Methods, or Discussion.

Response: We appreciate the reviewer's suggestion and have now expanded the details in the MIRAGE document (**Supplementary File 3**) to provide additional information about the experimental conditions tested during the optimisation studies. These details include different fixation and labelling conditions for various bacterial strains, the selection of microarray slides, printing volumes and glycan probe concentrations assessed during microarray construction. We have also edited the results and discussion regarding glycan concentration on Microarray set 2a/b for the study of binding to neutral and acidic glycans. Further information regarding preparation of bacteria for screening is also included in the methods section and **Extended figures 2 and 4**.

Methods

1.12 The authors mention profiling the arrays with a bunch of antibodies and lectins in the experimental section. However, there is no mention of this in the main text. What were the results? Were they as expected? For example, their anti-LeY antibody bound well to LNnFPI-AEAB, H Tetra T2-Lac-PTPA, and Hexa T1-T2-Lac-PTPA – was that expected? There are some pretty big differences between the two BG-A antibodies, T36 and Z2A. was that expected? The LeA antibody binds sialyl-LeA and sulfo-LeA – is that expected?

Response: Quality control (QC) of microarrays involved lectin and antibody binding analysis. Considering the manuscript length limitations, we did not include a detailed description of these results in the original manuscript. We have now included a description of results from the QC analyses including a few unpredicted (or not previously reported) findings with some commercial antibodies in the MIRAGE document under heading "Glycan identification and quality control" (**Supplementary File 3**), and in the table footnotes in **Supplementary File 2** summarising the QC data of each subarray. We have also added a sentence referring to the QC data in the results section (page 4, lines 111-117).

We greatly appreciate the careful review of the antibody data by this reviewer. Upon thoroughly re-examining the raw microarray data, we identified a shift had occurred during quantitation of the anti-LeY experiment on Microarray set 2b,, leading to the

*unexpected binding observed with this antibody. This error has now been corrected, and the revised results are provided in **Supplementary File 2**.*

1.13 Also, how do the results for the lectins and antibodies help with interpreting results. For example, they talk about bacteria not binding blood group antigens (page 6) – were these bound by the anti-blood group antibodies? This information would confirm that it's a real effect and not a printing issue.

Response: *As described above, we have included quality control data for all probes used in the study in **Supplementary File 2**. As alluded to by the reviewer, the purpose of undertaking these experiments was indeed to confirm that all glycan probes were properly immobilized in each array for bacterial binding analysis. As an example, none of the 20 short glycan probes with the sp linker (**Supplementary File 2_Microarray set 3, probes #91-110**) bound bacteria. However, good binding with lectins and antibodies was observed indicating that these probes were appropriately immobilised and presented on the arrays. In contrast, related glycans with similar blood group terminating epitopes presented in longer chains, were selectively bound by bacteria as well as lectins and antibodies. This has now been clarified in the text (lines 149-152).*

Characterization data.

1.14 The authors state that many of the compounds were characterized by MS. However, a number of others don't mention any characterization data, such as AEAB, Ser/Thr core 1 probes. The authors should provide at least some characterization data for all new compounds described in this paper. Ideally, this would include MS and NMR. however, I realize they may have only gotten small amounts of many of them, making NMR difficult. Where possible, some data for composition and purity should be included for the compounds. Additionally, they should include copies of the actual data (e.g. MS spectra) in the supporting information. In cases where this is not feasible, they should acknowledge that the purity and/or composition has not been determined (perhaps in Extended Table 1).

Response: *As requested, we have now included comprehensive MS data for the newly prepared PTPA (**Supplementary File 1_Table 1**) and AEAB (**Supplementary File 1_Table 2**) probes. Additionally, we have included mass spectra for selected probes of other types of glycans in seven new figures (**Supplementary File 1_Figure X1 to Figure X7**). These figures also include HPLC profiles of final purification steps and some additional chromatograms indicating the high purities obtained, together with 2D-NMR HSQC spectra from two selected HMOs, where sufficient amounts of material were available. These additional data demonstrate the quality of the glycan probes used in the study.*

1.15 I had some difficulty figuring out how many bacteria were incubated on the array

and what volume they were in. From what I can tell, they bacteria were grown to an OD of 1.0 and then pelleted. What volume of culture was pelleted (or how many total cells were pelleted)? What volume where they resuspended in?

Response: *A total of 100 μ L of fluorescently labelled bacterial suspensions at OD=1 was applied to the incubation chamber. This is described in the methods section under heading "Bacterial binding analyses".*

1.16 For the SaHyal_767 protein expression, the authors need to include the exact nucleotide sequence used for the expression. For example, was it codon optimized? They should also include the final yield from a stated volume, so that others would know how much to expect if they repeated this work.

Response: *As requested, we have now included a new figure in **Supplementary File 10** detailing the nucleotide sequence of the gene encoding the SaHyal_767 enzyme, which was codon optimized for expression in *E. coli* (by GenScript). We have also clarified the gene sequence of this modular enzyme and the protein yield in the methods section.*

1.17 In the Fluorescent labeling section, please add the volume of PBS used.

Response: *This has been added to the methods section as requested.*

1.18 In the labeling of fixed bacteria, please add the volume of of HBSS used to resuspend the cells before staining.

Response: *This has now been added to the methods section as requested.*

Reviewer #2

We appreciate Reviewer #2's thoughtful and positive feedback on our study, and the recognition of the importance of glycan arrays in identifying bacterial glycan-binding partners and the strength of our follow-up binding assays with *in vitro*-grown vaginal epithelial cells. We are also pleased that this reviewer found the flow cytometry experiments well-designed and supportive of our conclusions. The acknowledgment of the significance of our findings, particularly the role of CSC in *Lactobacillus crispatus* binding, is greatly appreciated.

This reviewer seems to have a major concern regarding the use of HEPES buffer for experiments at pH 4: "HEPES, which is used in the binding buffer bacteria are resuspended in, does NOT have sufficient buffering capacity at pH 4 to make the conclusions about pH that the manuscript is putting forward (pH dependence is concluded throughout based on these methods)." "HEPES: It is very concerning that the authors seem to be stating (line 618) that they are using HEPES "at pH 7.5 or pH 4" as the "binding buffer". HEPES is not an effective buffer at pH 4. It does not have to be pH 4 to place into biological context (see next comment), but this error must be corrected!"

Response: *We appreciate the reviewer's concern. To the best of our knowledge, HEPES has two pKas, pKa1 (25 °C)= 7.5 and pKa2 (25 °C)= 3 and thus, while mainly used for buffering at neutral to slightly alkaline pH (6.8–8.2), can also be used for low pH, although the latter is often considered suboptimal for most applications. Our rationale for using HEPES at both pH conditions was to minimise changes in ion composition during bacterial binding events. Moreover, we carefully monitored pH throughout the microarray analyses. As detailed in the methods section " Bacterial binding analyses" (Page 24) HBS buffer (pH7) was prepared and kept for no more than 4 weeks at room temperature. HBS at pH4 was prepared immediately before use by adjusting the pH with HCl. The pH of the bacterial suspensions used in the microarray binding assays was monitored before and after experiments using pH strips (Fisherbrand™ 92110.204231) and was shown to remain constant.*

*To address any doubts regarding the pH dependence of bacterial binding to GAGs, we have carried out GAG binding assays with selected bacterial strains using acetate buffer and demonstrated that vaginal bacteria binding to CS and Heparin oligo- and polysaccharides is indeed pH dependent (see new **Supplementary Figure 8**).*

MAJOR COMMENTS:

2.1 The focus on pH in the vagina is relevant. The methods used can have great impact on how the findings are interpreted. Please provide more information about how buffers were prepared and used.

PBS: There are many ways to prepare PBS. The authors here use "PBS" at two different pHs (4 and 7) to make conclusions about "pH-dependent" binding throughout the paper. It is of key importance to clearly describe these methods. How was the buffer prepared and pH-titrated? PBS has two pH ranges where it can act as an effective buffer. Did the authors ensure that buffering capacity was maintained during the binding assays? There are wash and resuspension steps should be better described too.

Response: *PBS buffer was commercially sourced (Sigma (D8537)). For flow cytometry experiments, PBS was adjusted to pH4 with HCl, which remained stable through the experiment as assessed using a pH meter.*

2.2 The pH ranges associated with the Lactobacillus-dominant microbiotas is around pH 4.0-4.4 for Lactobacillus-dominant microbiotas (CSTI and CSTIII) and pH 5.3-5.5 for BV/CST IV microbiotas (Ravel, 2011, PNAS). pH 4 is relevant, but using pH 7 limits the conclusions and biological interpretations one can make about how these glycan-binding interactions might operate in the vagina these major community state types. Though not physiologically impossible, pH 7 would be an outlier in BV, where the average is ~ pH 5.5. In light of the issues raised above about pH and buffering capacity, it would be good to take another stab at this.

Response: *While the pH range within the vagina is indeed between 4 - 4.5 for most women, the pH of the cervical region between 6.7-7.2 and in the upper uterine cavity, between 7.5-8.0) (Lykke et al. Pathogens, 2021)⁵. Several of the pathogens that we have examined in our study, including *F. nucleatum* and *S. agalactiae*, cause ascending intrauterine infections in pregnant women resulting in colonisation of the fetal membranes. Thus, our findings do have important biological and clinical relevance, particularly given that some of the specific glycan targets of these bacteria at pH7 (e.g. HA and CSA), are enriched in the cervix and in fetal membranes during pregnancy. We have amended the manuscript to provide further justification for the pH range investigated in our study and to highlight the physiological relevance (see lines 143-145 and 344-348).*

2.3 Figure 1A needs to be a lot bigger – this is a key figure that enables the reader to see the scope of your work!. The rest of this figure is not necessary to show in the main body of the paper. This validation of the array can go to the extended figures. It would be better to provide examples of the classes of glycans tested on the array (like Figure 3 SNFG glycans; this is very nice!)

Response: *We thank the reviewer for these suggestions. We have produced a new version of **Figure 1**, which now includes examples of the classes of glycans tested on the*

array.

2.4 I find Extended Figure 2 to be much more informative than Figure 2 and feel that these figures should be switched (Extended Figure 2 should be shown along with the main text). The highlighted boxes labeled with the different types of glycans that is shown in current Fig 2 is nice. Please put this in both figures.

Response: *As suggested, we have swapped **Figure 2** with **Extended Figure 2**, and have included in both the highlighted boxes labelled with the different types of glycans.*

2.5 Binding assays are at ambient temperature and with live bacteria. How can you be sure the bacteria are binding to the glycan presented in the array? In the presence of bacterial exoglycosidase activities, it could be reasoned that some bacteria remove outer monosaccharides to bind inner glycan epitopes. For example, the authors find that virtually all of the bacteria bind to sialylated glycans in multiple presentations. Some of these bacteria (e.g. Gardnerella) have also been found to express glycosidase activities and might bind underlying glycan structures. This should be considered and the arguments presented for how the array technology might be able to tease this apart.

Response: *The reviewer raises an interesting and important point, and we agree that modification of the glycans by live bacteria may occur during the binding assays. The experimental design and the comprehensive set of probes used on the arrays allowed us to detect bacterial binding to both the "putative substrate" and the modified glycan. Using the example provided by the reviewer, we observed binding of Gardnerella to sialylated glycans but not to matched galactose terminating glycan probes. Similar reasoning can be applied to the other strains tested in the study. We have included additional discussion on this point (lines 321-334).*

*Regarding the reviewer's comment about "multiple presentations", we understand they are referring to the presentation of glycans. To this point, we observed specific binding by *F. nucleatum* to α 2,6-sialic acid terminating glycans but not to α 2,3-sialic acid terminating glycans. While other gram-positive bacteria do bind α 2,3 and α 2,6-sialic acid terminating glycans, this is only observed when these glycans are presented on long polyLacNAc chains and not on short core-1 O-glycans.*

2.6 It is not clear that the data support the vague interpretation in the abstract and elsewhere that glycan binding signatures of opportunistic pathogens were "highly distinct" from commensals.

First, regarding the definition of "commensals": In the abstract, the authors refer to *Lactobacillus iners* and *Gardnerella* as "commensals", but by definition commensals do

not cause harm. There are a number of publications using tissue and animal models showing that both microbes express cytolytic toxins (inertolysin, vaginolysin) capable of lysing human cells. This seems like a strong argument against the idea that these microbes exist peacefully without causing harm to their host. This is further supported by health problems that have been associated with the community state types where these taxa are dominant (in comparison to *L. crispatus*).

If the authors plan to say glycan binding was “highly distinct” from one group of bacteria to another, please adequately justify the categorization of bacteria and also summarize the findings that support such a statement or make the statement more specific.

Response: *We appreciate the reviewer highlighting a lack of clarity around definitions for the categorisation of bacteria tested in the study. These definitions are challenging for the vaginal niche where major variation in host response and tolerance to different bacteria is observed. For example, while both Lactobacillus iners and Gardnerella species can produce cytolytic toxins, healthy, asymptomatic women are often colonised with these species and high relative abundance of these species in the niche does not always cause inflammation and pathology.*

For the sake of clarity, we now consistently refer to Lactobacillus species as ‘commensals’, with the caveat that L. iners is associated with increased risk of transition to a high-diversity composition often enriched by ‘opportunistic, potentially pathogenic’ species. These include G. vaginalis, S. agalactiae, F. nucleatum and E. coli, which under certain circumstances can cause acute infections, yet can also be detected in the vagina of asymptomatic women (lines 63-69).

2.7 Likewise at the end of the Introduction, the authors conclude that “our results reveal specific and distinctive glycan binding profiles of vaginal commensals and pathogens”. It is not clear what specific and distinctive results support this statement. Except for the couple of examples already known (*E. coli* FimH and mannose, *F. nucleatum* Fap2), it seems like there is actually more in common across the microbes than there are differences. Please make a concluding statement that is more specific to the findings.

Response: *We agree with the reviewer that our concluding remarks in the introduction could have been more accurate and have re-written this as requested.*

Context and Physiological Relevance of bacterial-glycan interactions:

2.8 Figure 1 (*E. coli*) and Figure 3 (*F. nucleatum*) are relevant to validate the arrays because of known binding activities between these bacteria and relevant glycans. These are nice figures. However, it is surprising given the availability of FimH mutants

in UPEC and Fap2 mutants in *F. nucleatum* that the authors did not obtain and use these mutants to further establish the specificity of binding both in terms of the glycans and the lectins that recognize them. There is speculation in the discussion that additional binding activities seen for *F. nucleatum* is "possibly mediated by Fap2 or adhesins yet to be identified." This is testable and would provide a substantial advance by revealing Fap-2-independent processes not yet discovered.

Response: *We thank the reviewer for this suggestion. We have now investigated glycan binding by the Fap2 deletion mutant of F. nucleatum 23726 and included the new data in the results (Figure 4B and Results, lines 173-176). These findings represent hitherto unknown binding activities between Fusobacterium and specific sequence-defined glycans (galactose and sialic acid terminating). The discussion has been amended to provide context and interpretation of the importance of these findings (lines 356-366).*

2.9 The authors conclude (line 266) that the EM data shown for *E. coli* "provided a structural explanation for the binding to high mannose structures." This statement seems disingenuous, given the large literature describing interactions of UPEC FimH-adhesin that binds mannose structures. The "structural explanation" claimed by the authors pales in comparison to what is known already in the literature.

Response: *We agree that our choice of wording here was not accurate. We have now rephrased the sentence to state that "the abnormally short and reduced number of cellular appendices in E. coli 900 is likely to be abrogating oligo-mannose binding in this strain" in the discussion (lines 282-284).*

2.10 There is a single weighty sentence spanning lines 186 to 180. However, it seems to be missing information and is not understandable. The sentence states bacterial binding was strongest "on covalent arrays" but in comparison to what? It is not clear why the authors conclude that this suggests "that presentation of glycans within the chemical environment is an important mediator of interactions." Are the authors referring to background binding to the chemical linkers used for some of the glycans? If so, this needs to be stated more clearly.

Response: *The reviewer is correct – this sentence contains missing information and is unclear. Only covalent arrays were used in this work. No background binding was observed to any of the linkers used. We have now revised the sentence to read "With the exception of F. nucleatum, we observe that most strains bound to neutral glycans derivatised with the AEAB (N-(aminoethyl)-2-aminobenzamide) linker (Fig. 3), suggesting that presentation and the chemical environment around the glycan, are likely to be important for whole bacterial binding to glycans^{6, 7}. Again, this effect was not observed in antibody and plant lectin binding (Supplementary File 2)" (lines 314-319).*

2.11 Heparin did not get a proper introduction in the manuscript and is presented as a finding without any context. It is not clear why heparin would be relevant to the vaginal mucosa.

Re: "Non sulfated" chondroitin sulfate doesn't seem relevant (control) to lectin-glycan interactions between host and vaginal bacteria – aren't these precursors inside cells that are used to build CS?

Response: *As described from line 210, Heparin was used a proxy for highly sulphated domain of heparan sulphate (HS), which is expressed in the lower reproductive tract and the endometrium.*

Non-sulphated chondroitin sulphate (CS-0S) was used as a negative control probe for sulphated CS in consideration of the possibility that the degree of sulphation is an important factor in lectin-GAG interactions. Moreover, under-sulphated CSA with non-sulphated CS disaccharide unit is present in human placenta and has important biological roles (e.g.: Achur et al. J. Biol. Chem, 2000; Chai et al. J. Biol. Chem, 2002)^{8, 9}.

2.12 The final sentence of the abstract "This study highlights glycans as key mediators of vaginal bacterial binding events and targets for microbiome modulating therapeutic strategies," is overly broad and needs to be qualified better. There is much to do to ascertain the key binding events for vaginal bacterial binding and how it participates in colonization or pathogenesis. I do NOT think this study succeeds in highlighting "glycans as targets for microbiome modulating strategies". This is a huge leap.

Response: *In consideration of the reviewer's comments, we have modified the last sentence in the abstract as follows, "This study highlights glycans as key mediators of vaginal bacterial binding events involved in reproductive health and disease".*

2.13 Can the authors clarify exactly where they used fixed bacteria prior to some of the binding assays? Is it only Figure 1 and extended Figure 1 that used fixed bacteria? If not and this strategy is sprinkled throughout the paper, can the authors please concisely state in the figure legends whether bacteria are fixed and labeled prior to binding or whether they undergo the live labeling strategy?

Response: *As requested, we have now specified in the methods and figure legends that only E. coli and F. nucleatum cultures were fixed prior to undertaking binding assays.*

2.14 The data presented in Figure 5B seems problematic. The histogram on the left shows HUGE binding at pH7 by L. crispatus with a peak intensity around 10^3 but the right side shows data in which the signal seems to be maxing out a little over 10^2 .

Response: We thank the reviewer for highlighting this error. The wrong histogram was indeed inserted for *L. crispatus* pH7 in Figure 5B. We have now replaced this with a representative histogram generated from one of the three repeats shown in the bar graph on the right of the figure.

2.15 Extended Figure 6. It seems problematic that the methods for panel C (*G. vaginalis* 775) compare the pH 4 and pH 7 conditions using different MOIs. How do the authors justify that they used MOI of 100 only for the *G. vaginalis* pH 7 condition?

Response: A ratio of bacterial particles to cells or MOI (Multiplicity of Infection)=100 was used only for *G. vaginalis* at pH 4. *G. vaginalis* binds poorly the vaginal epithelial cells at pH 4 and very low binding was observed at MOI 20. We therefore increased the MOI for *G. vaginalis* to MOI, which allowed for observation of bacterial binding and inhibitory effects. For all other conditions, an MOI 20 was used. This is now clearly explained in the methods section under "Assessment of VK2-glycan and bacteria binding by flow cytometry".

2.16 The discussion refers to prior work from this group (ref 41 line 277) that made computational predictions of possible lectins from vaginal bacteria. It seems the current dataset does not bear out those predictions (?). I would welcome a paragraph about this in the discussion.

Response: The work with the comparative genomics study (ref 42 in manuscript) revealed a putative (*de novo*) wider repertoire of GBPs in vaginal pathogens and pathobiont compared to commensal species. We observed additional glycan binding activities in pathobionts over those observed in the commensal *L. crispatus* including high mannose binding by *E. coli*, HA binding by GBS and Gal binding by *F. nucleatum*. These findings are all in accordance with our previous report. However, detecting glycan binding activities by secreted glycan binding proteins or enzymatically active glycanases that are transient, might be more challenging and reflects a limitation of the glycan microarray system. Further comparison of the findings is challenging given that the expression of glycan binding proteins will reflect only a subset of those identified *de novo* and will be subject to change depending on the growth conditions. In this initial study we have used rich growth medias for the bacteria, and we have studied each strain separately. Follow up studies with complex bacterial populations and or in different culture conditions might identify further glycan binding activities. These points are now included in our discussion (lines 321-334).

MINOR:

First sentence of the abstract uses causal language, "increases risk" should be "is associated with higher risk of"

Response: *Corrected.*

HEPES buffer should be in all caps, line 618 and possibly other places, please find and replace

Response: *Corrected.*

Line 575; please specify a maximum time that labeled bacteria were stored as stated prior to use.

Response: *This is now specified as requested.*

Extended figure 5 Title should say "bound by *F. nucleatum*" not "bound by bacterial of the vaginal microbiome", since only *F. nucleatum* data is presented in the figure.

Response: *Corrected.*

There are two panel (B)s listed in the legend for Figure four, the second one should be (C).

Response: *Corrected.*

Line 240– should refer to Fig 5B not 6B

Response: *Corrected.*

Line 246 should refer to Fig 5C not 6C

Response: *Corrected.*

Please remove "recently" from line 320 and provide an earlier reference. This was demonstrated a long time ago.

Response: *As requested, we have removed the term "recently" from this sentence and provided earlier references supporting the relationship between *F. nucleatum* colonisation, PPRM and preterm delivery.*

Line 610. The methods mention using Siglec15-Fc reagents, but the manuscript does not seem to include this.

Response: *We used Siglec15-Fc to detect Lstb in Microarray set 2. Please see **Supplementary File 2***

The statement line 241-242, "S. agalactiae and L. crispatus rarely coexist in the vaginal niche" is incorrect and not supported by the paper cited, which does not name taxa to the species level.

Response: *We thank the reviewer for pointing this out. We mistakenly included a reference that does not support the statement that S. agalactiae and L. crispatus rarely coexist in the vaginal niche. We have now updated the text and provided appropriate supporting references.*

Reviewer #3

3.1 This is very nice descriptive work detailing glycan interactions with some key vaginal bacteria, GBS and pathobionts. The significance of the work is centered on preterm birth, PPRM and neonatal outcomes. As such, the title should be revised to reflect the focus of the paper and the discussion. In terms of the clinical relevance and translation of the work, the authors suggest in the last sentence of the discussion that *L.crispatus* based live bio therapeutics for preventing preterm birth, but this is not a new or original finding resulting from this work. It is however another aspect of *L. crispatus* dominant microbiomes that could suggest a potential rationale for outcompeting pathogenic bacteria. Unfortunately, the authors do not demonstrate that phenomenon in the work. Additional evidence is needed

Response: *We thank the reviewer for their positive comments. As suggested, we have now revised the title of our paper to "Identification and characterisation of vaginal bacteria-glycan interactions implicated in reproductive health and pregnancy outcomes". We feel that this more accurately reflects the interpretation and contextualisation of the findings presented. However, we would like to note that although the bacteria examined in our study are indeed associated with adverse pregnancy outcomes, these species are also more broadly implicated in other areas of reproductive tract health and disease including bacterial vaginosis, risk of STI acquisition and HPV infection and progression of cervical cancer. We have now substantially edited the discussion to include the broader significance of our findings (lines 424-444).*

*We agree with the reviewer that the idea of using *L. crispatus* as a biotherapeutic is not new and have rephrased this accordingly.*

3.2 Abstract/Intro: It is not clear why the specific bacterial genus/species were selected for the study? For example, Prevotella is a key vaginal bacteria and known to interact with glycans. .why wasn't that bacteria studied?

There is no description or rationale outlined for the specific bacteria and species that were included.

Response: *As alluded to in the previous reply, bacteria selected for study were largely based on their association with risk of adverse pregnancy outcomes and more broadly, reproductive health and disease. Further, we limited our selection of strains to those derived from the reproductive tract. Limitations of resources also meant that not all species and strains that colonise the vagina could be covered in the current study. We have now modified the text in the introduction and results sections to provide a rationale for the selection of bacteria (lines 63-69 and 139-141).*

3.3 There has been a restructuring of *G.vaginalis* and other key species identified..how does that relate to the work presented herein? Why did the authors select the *G.vaginalis* that was included in the study? Suggest adding a rationale for the selection of these bacteria, species, # of strains tested as strain specific differences exist.

Response: *The recent expansion of Gardnerella vaginalis into several different unique genomospecies is indeed of relevance to our work^{10, 11}. All strains used in our study were confirmed to be G. vaginalis using metagenomics approaches (Thomas-White, et al., 2018) and MALDI-TOF. While the rationale of selecting these strains has now been provided (see point 3.2 above), our findings highlight certain isolate-to-isolate variability on glycan binding.*

Subspecies and strain to strain variability of G. vaginalis have been shown to differ in terms of virulence factors expressed, biofilm formation capacity and/or variability on the mucosal immune responses elicited¹². Such variations could underlie differing functional abilities including glycan binding as observed in this study, with exposure of different lectins, capsule or cell wall components. Consistent with these findings, specific sub-species or strains of G. vaginalis have recently been implicated in increased risk of preterm birth^{13, 14}. Despite this, some consistent binding to certain glycans like H type 1 containing human milk oligosaccharides was observed in our study. These findings are in agreement with the variability observed in the predicted carbohydrate binding protein repertoire of vaginal pathogen and commensal species¹⁵ reported before and highlight the importance of examining the relationship between strain-level resolution of the vaginal microbiota, glycan binding and use, host inflammatory response and clinical outcomes. This information is now included in the discussion (lines 376-385 and 327-334).

3.5 Results: The data analysis and interpretation are clear and methodology is sound for the glycan analysis. More glycan interaction work elucidating a mechanism is needed. It is not clear what is gained by the in vitro experiments. Do the authors anticipate that culturing E6/E7 immortalized cells in monolayer would recapitulate the in vivo representation of glycan presentation? This was not acknowledged in the manuscript.

Response: *We have undertaken several studies of GAG expression in the VK2 cells before performing bacterial binding experiments. We observed high expression of heparan sulphate and chondroitin sulphate, but not hyaluronic acid, on these cells as studied by flow cytometry with specific GAG binding antibodies followed by specific hydrolases treatment. These results, presented in **Figure 6** and **Extended Figure 6**, align well with what has been reported so far in the literature regarding GAG expression in the lower reproductive tract¹⁶⁻¹⁸. Collectively, our in vitro findings provide evidence that the adhesion of key vaginal bacteria to vaginal epithelial cells is partially mediated by*

chondroitin sulphate, and that S. agalactiae binding to the chondroitin sulphate C oligosaccharide, can be inhibited by L. crispatus.

We acknowledge that these findings are limited to an in vitro context and have accordingly amended the text (see line 412).

3.6 Mucin expression is impacted by polarization of epithelial cells and monolayers are not polarized and do not have the same levels of expression. How does this relate to glycan expression?

Response: *This is an interesting point and indeed, important work by K. Ribbeck and others has shown that glycans on mucins can modulate the virulence of candida in 3D cell models. However, the primary focus of the in vitro cell binding study conducted in our study was to assess bacterial binding to GAGs expressed on proteoglycans on the cell surface, rather than to mucin O-glycans. As per the previous response, we have amended the text to highlight the limitation of our in vitro data.*

3.7 The methods are detailed to allow for reproduction. The discussion could be further enhanced with more of a clinical/translational perspective. Again it is focused primarily on obstetric health and more specifically preterm birth so the title should reflect that focus. What are the significance of the findings? Is this data providing more of a compelling argument for L.crispatus role in full term pregnancies? This was not clear beyond the work that has been published on blood groups previously and a suggestion towards L.crispatus based biotherapeutics for microbiome modulation. The work is solid, but overall descriptive and I was left wanting more of the mechanism, clinical significance and impact on the field in this section. What is the future application?

Response: *We have now substantially altered the discussion of the paper to more clearly describe the potential translation of these findings to a clinical context. This includes highlighting more clearly how the results provide new insight into the potential mechanism by which L. crispatus can prevent and/or displace potentially pathogenic bacteria from the vaginal niche. These findings provide further evidence that L. crispatus has an important role in promoting optimal vaginal health including in pregnancy, where preventing adhesion of pathogens would reduce risk of ascending infection and consequently, adverse outcomes such as preterm birth. As suggested, the title of the paper has been revised to better reflect the focus of the discussion.*

References

1. Wu, G. *et al.* Glycomics of cervicovaginal fluid from women at risk of preterm birth reveals immuno-regulatory epitopes that are hallmarks of cancer and viral glycosylation. (2024).
2. Wu, G. *et al.* N-glycosylation of cervicovaginal fluid reflects microbial community, immune activity, and pregnancy status. *Scientific reports* **12**, 16948 (2022).
3. Kundu, S. *et al.* Secretor status is a modifier of vaginal microbiota-associated preterm birth risk. *Microbial genomics* **10** (2024).
4. Mirmonsef, P. *et al.* Free glycogen in vaginal fluids is associated with *Lactobacillus* colonization and low vaginal pH. *PloS one* **9** (2014).
5. Lykke, M. *et al.* Vaginal, Cervical and Uterine pH in Women with Normal and Abnormal Vaginal Microbiota. *Pathogens (Basel, Switzerland)* **10**, 90 (2021).
6. Feizi, T. Angling for recognition. *Current biology : CB* **2**, 185-187 (1992).
7. Grant, O., Smith, H., Firsova, D., Fadda, E. & Woods, R. Presentation, presentation, presentation! Molecular-level insight into linker effects on glycan array screening data. *Glycobiology* **24**, 17-25 (2014).
8. Chai, W., Beeson, J. & Lawson, A. The structural motif in chondroitin sulfate for adhesion of *Plasmodium falciparum*-infected erythrocytes comprises disaccharide units of 4-O-sulfated and non-sulfated N-acetylgalactosamine linked to glucuronic acid. *The Journal of biological chemistry* **277**, 22438-22446 (2002).
9. Achur, R., Valiyaveetil, M., Alkhalil, A., Ockenhouse, C. & Gowda, D. Characterization of proteoglycans of human placenta and identification of unique chondroitin sulfate proteoglycans of the intervillous spaces that mediate the adherence of *Plasmodium falciparum*-infected erythrocytes to the placenta. *The Journal of biological chemistry* **275** (2000).
10. Munch, M. *et al.* Gardnerella Species and Their Association With Bacterial Vaginosis. *The Journal of infectious diseases* **230** (2024).
11. Vanechoutte, M. *et al.* Emended description of *Gardnerella vaginalis* and description of *Gardnerella leopoldii* sp. nov., *Gardnerella pottii* sp. nov. and *Gardnerella swidsinskii* sp. nov., with delineation of 13 genomic species within the genus *Gardnerella*. *International journal of systematic and evolutionary microbiology* **69** (2019).
12. Shvartsman, E., Hill, J., Sandstrom, P. & MacDonald, K. Gardnerella Revisited: Species Heterogeneity, Virulence Factors, Mucosal Immune Responses, and Contributions to Bacterial Vaginosis. *Infection and immunity* **91** (2023).
13. Pace, R. *et al.* Complex species and strain ecology of the vaginal microbiome from pregnancy to postpartum and association with preterm birth. *Med (New York, N.Y.)* **2** (2021).
14. Callahan, B. *et al.* Replication and refinement of a vaginal microbial signature of preterm birth in two racially distinct cohorts of US women. *Proceedings of the National Academy of Sciences of the United States of America* **114**, 9966-9971 (2017).
15. Bonnardel, F. *et al.* Proteome-wide prediction of bacterial carbohydrate-binding proteins as a tool for understanding commensal and pathogen colonisation of the vaginal microbiome. *NPJ biofilms and microbiomes* **7**, 49 (2021).

16. Akgul, Y., Holt, R., Mummert, M., Word, A. & Mahendroo, M. Dynamic changes in cervical glycosaminoglycan composition during normal pregnancy and preterm birth. *Endocrinology* **153**, 3493-3503 (2012).
17. Nasciutti, L. *et al.* Distribution of chondroitin sulfate in human endometrium. *Micron (Oxford, England : 1993)* **37**, 544-550 (2006).
18. Mahendroo, M. Cervical hyaluronan biology in pregnancy, parturition and preterm birth. *Matrix biology : journal of the International Society for Matrix Biology* **78-79**, 24-31 (2019).